# UltraPixel: Advancing Ultra-High-Resolution Image Synthesis to New Peaks

**Jingjing Ren**[1][*], **Wenbo Li**[2][*], **Haoyu Chen**[1], **Renjing Pei**[2], **Bin Shao**[2],
**Yong Guo**[3], **Long Peng**[2], **Fenglong Song**[2], **Lei Zhu**[1,4][†]
[1]HKUST (Guangzhou)  [2]Huawei Noah's Ark Lab  [3]MPI  [4]HKUST
Project page: https://jingjingrenabc.github.io/ultrapixel

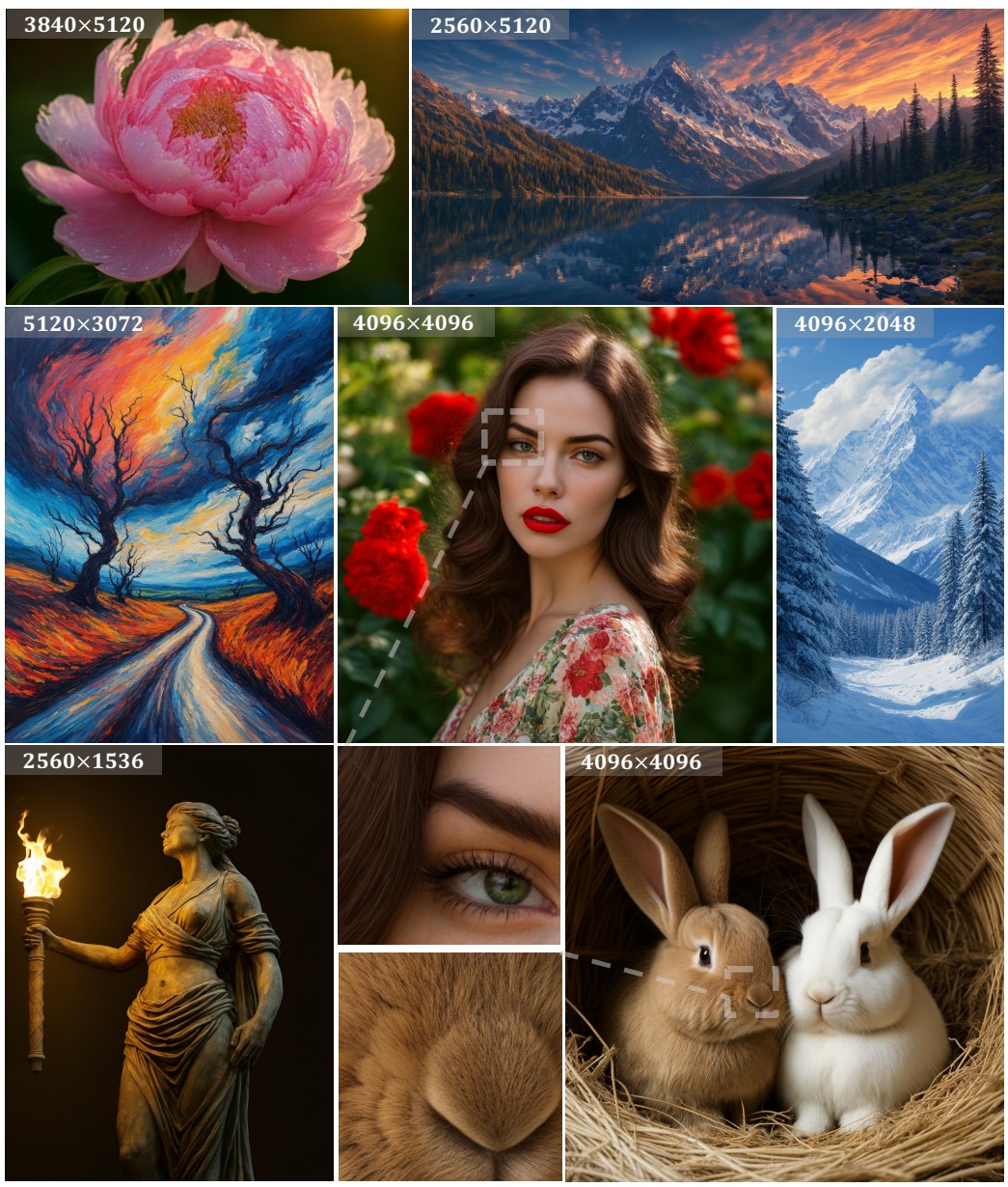

Figure 1: The proposed UltraPixel creates highly photo-realistic and detail-rich images at various resolutions. Best viewed zoomed in. **All image prompts in this paper are listed in the appendix.**

[*]Joint first authors
[†]Corresponding author

38th Conference on Neural Information Processing Systems (NeurIPS 2024).

## Abstract

Ultra-high-resolution image generation poses great challenges, such as increased semantic planning complexity and detail synthesis difficulties, alongside substantial training resource demands. We present UltraPixel, a novel architecture utilizing cascade diffusion models to generate high-quality images at multiple resolutions (*e.g.*, 1K to 6K) within a single model, while maintaining computational efficiency. UltraPixel leverages semantics-rich representations of lower-resolution images in the later denoising stage to guide the whole generation of highly detailed high-resolution images, significantly reducing complexity. Furthermore, we introduce implicit neural representations for continuous upsampling and scale-aware normalization layers adaptable to various resolutions. Notably, both low- and high-resolution processes are performed in the most compact space, sharing the majority of parameters with less than 3% additional parameters for high-resolution outputs, largely enhancing training and inference efficiency. Our model achieves fast training with reduced data requirements, producing photo-realistic high-resolution images and demonstrating state-of-the-art performance in extensive experiments.

# 1 Introduction

Recent advancements in text-to-image (T2I) models, *e.g.*, Imagen [42], SDXL [37], PixArt-$\alpha$ [4], and Würstchen [36], have demonstrated impressive capabilities in producing high-quality images, enriching a broad spectrum of applications. Concurrently, the demand for high-resolution images has surged due to advanced display technologies and the necessity for detailed visuals in professional fields like digital art. There is a great need for generating aesthetically pleasing images in ultra-high resolutions, such as 4K or 8K, in this domain.

While popular T2I models [37, 4, 36] excel in generating images up to $1024 \times 1024$ resolution, they encounter great difficulties in scaling to higher resolutions. To address this, training-free methods have been proposed that modify the network structure [15, 22] or adjust the inference strategy [1, 12, 21] to produce higher-resolution images. However, these methods often suffer from instability, resulting in artifacts such as small object repetition, overly smooth content, or unreasonable details. Additionally, they frequently require long inference time [12, 14, 21] and manual parameter adjustments [15, 14, 12] for different resolutions, hindering their practical applications. Recent efforts have focused on training models specifically for high resolutions, such as ResAdapter [6] for $2048 \times 2048$ pixels and PixArt-$\Sigma$ [4] for $2880 \times 2880$. Despite these improvements, the resolution and quality of generated images remain limited, with models optimized for specific resolutions only.

Training models for ultra-high-resolution image generation presents significant challenges. These models must manage complex semantic planning and detail synthesis while handling increased computational loads and memory demands. Existing techniques, such as key-value compression [3] in attention [40, 11, 38, 39, 35] and fine-tuning a small number of parameters [6], often yield suboptimal results and hinder scalability to higher resolutions. Thus, a computationally efficient method supporting high-quality detail generation is necessary. We meticulously review current T2I models and identify the cascade model [36] as particularly suitable for ultra-high-resolution image generation. Utilizing a cascaded decoding strategy that combines diffusion and variational autoencoder (VAE), this approach achieves a 42:1 compression ratio, enabling a more compact feature representation. Additionally, the cascade decoder can process features at various resolutions, as illustrated in Section A in the appendix. This capability inspires us to generate higher-resolution representations within its most compact space, thereby enhancing both training and inference efficiency. However, directly performing semantic planning and detail synthesis at larger scales remains challenging. Due to the distribution gap across different resolutions (*i.e.*, scattered clusters in the t-SNE visualization in Figure 2), existing models struggle to produce visually pleasing and semantically coherent results. For example, they often result in overly dark images with unpleasant artifacts.

In this paper, we introduce UltraPixel, a high-quality ultra-high-resolution image generation method. By incorporating semantics-rich representations of low-resolution images in the later stage as guidance, our model comprehends the global semantic layout from the beginning, effectively fusing text information and focusing on detail refinement. The process operates in a compact space, with low- and high-resolution generation sharing the majority of parameters and requiring less than 3%

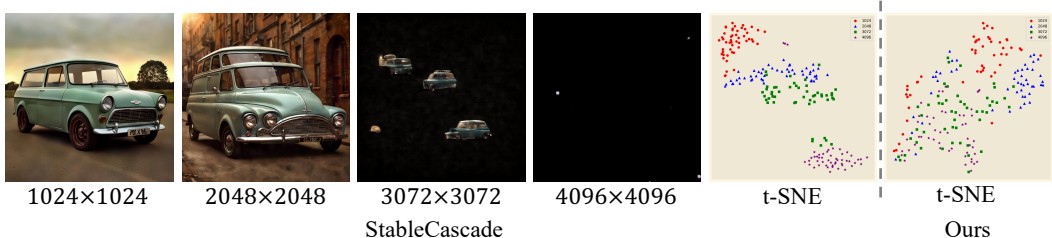

| 1024×1024 | 2048×2048 | 3072×3072 | 4096×4096 | t-SNE | t-SNE |

StableCascade

Ours

Figure 2: Illustration of feature distribution disparity across varying resolutions.

additional parameters for the high-resolution branch, ensuring high efficiency. Unlike conventional methods that necessitate separate parameters for different resolutions, our network accommodates varying resolutions and is highly resource-friendly. We achieve this by learning implicit neural representations to upscale low-resolution features, ensuring continuous guidance, and by developing scale-aware, learnable normalization layers to adapt to numerical differences across resolutions. Our model, trained on 1 million high-quality images of diverse sizes, demonstrates the capability to produce photo-realistic images at multiple resolutions (*e.g.*, from 1K to 6K with varying aspect ratios) efficiently in both training and inference phases. The image quality of our method is comparable to leading closed-source T2I commercial products, such as Midjourney V6 [32] and DALL·E 3 [34]. Moreover, we demonstrate the application of ControlNet [52] and personalization techniques [20] built upon our model, showcasing substantial advancements in this field.

## 2 Related Work

**Text-guided image synthesis.** Recently, denoising diffusion probabilistic models [45, 18] have refreshed image synthesis. Prominent text-guided generation models [37, 4, 3, 36, 9, 35, 29, 46, 42, 27] have demonstrated a remarkable ability to generate high-quality images. A common approach is to map raw image pixels into a more compact latent space, in which a denoising network is trained to learn the inverse diffusion process [4, 3, 37]. The use of variational autoencoders [23] has proven to be highly efficient and is crucial for high-resolution image synthesis [13, 41]. StableCascade [36] advances this approach by learning a more compact latent space, achieving a compression ratio of 42:1 and significantly enhancing training and inference efficiency. We build our method on StableCascade primarily due to its extremely compact latent space, which allows for the efficient generation of high-resolution images.

**High-resolution image synthesis.** Generating high-resolution images has become increasingly popular, yet most existing text-to-image (T2I) models struggle to generalize beyond their trained resolution. A straightforward approach is to generate an image at a base resolution and then upscale it using super-resolution methods [51, 10, 28, 48, 8]. However, this approach heavily depends on the quality of the initial low-resolution image and often fails to add sufficient details to produce high-quality high-resolution (HR) images. Researchers have proposed direct HR image generation as an alternative. Some training-free approaches [15, 12, 21, 1, 22, 53, 26] adjust inference strategies or network architectures for HR generation. For instance, patch-based diffusion [1, 26] employ a patch-wise inference and fusion strategy, while ScaleCrafter [15] modifies the dilation rate of convolutional blocks in the diffusion UNet [37, 41] based on the target resolution. Another method [22] adapts attention entropy in the attention layer of the denoising network according to feature resolutions. Approaches like Demofusion [12] and FouriScale [21] design progressive generation strategies, with FouriScale further introducing a patch fusion strategy from a frequency perspective.

Despite being training-free, these methods often produce higher-resolution images with noticeable artifacts, such as edge attenuation, repeated small objects, and semantic misalignment. To improve HR image quality, PixArt-sigma [3] and ResAdapter [6] fine-tune the base T2I model. However, their results are limited to $2880 \times 2880$ resolution and exhibit unsatisfied visual quality. Our method leverages the extremely compact latent space of StableCascade and introduces low-resolution (LR) semantic guidance for enhanced structure planning and detail synthesis. Consequently, our approach can generate images up to 6K resolution with high visual quality, overcoming the limitations of previous methods.

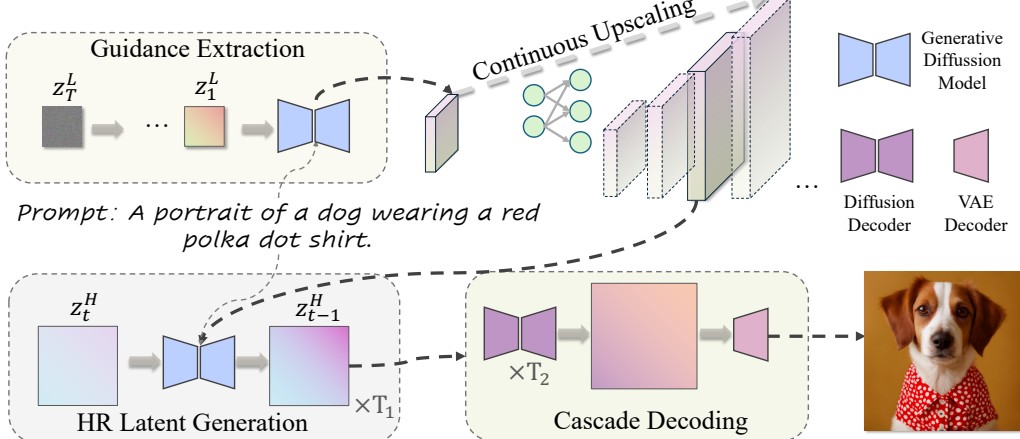

Figure 3: Method Overview. Initially, we extract guidance from the low-resolution (LR) image synthesis process and upscale it by learning an implicit neural representation. This upscaled guidance is then integrated into the high-resolution (HR) generation branch. The generated HR latent undergoes a cascade decoding process, ultimately producing a high-resolution image.

## 3 Method

Generating ultra-high-resolution images necessitates complex semantic planning and detail synthesis. We leverage the cascade architecture [36] for its highly compact latent space to streamline this process, as illustrated in Figure 3. Initially, we generate a low-resolution (LR) image and extract its inner features during synthesis as semantic and structural guidance for high-resolution (HR) generation. To enable our model to produce images at various resolutions, we learn implicit neural representations (INR) of LR and adapt them to different sizes continuously. With this guidance, the HR branch, aided by scale-aware normalization layers, generates multi-resolution latents. These latents then undergo a cascade diffusion and VAE decoding process, resulting in the final images. In Section 3.1, we detail the extraction and INR upscaling of LR guidance. Section 3.2 outlines strategies for fusing LR guidance and adapting our model to various resolutions.

### 3.1 Low-Resolution Guidance Generation

To address the challenges of high-resolution image synthesis, Previous studies [42, 17] have often employed a progressive strategy, initially generating a low-resolution image and then applying diffusion-based super-resolution techniques. Although this method improves image quality, the diffusion process in the pixel space remains resource-intensive. The cascade architecture [36], achieving a 42:1 compression ratio, offers a more efficient approach to this problem.

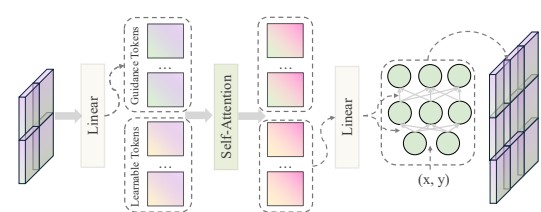

Figure 4: Illustration of continuous upscaling by implicit neural representation.

**Guidance extraction.** Instead of relying solely on the final low-resolution output, we introduce multi-level internal model representations of the low-resolution process to provide guidance. This strategy is inspired by evidence suggesting that representations within diffusion generative models encapsulate extensive semantic information [49, 2, 31]. To optimize training efficiency and stability, we leverage features in the later stage, which delineate clearer structures compared to earlier stages. This approach ensures that the high-resolution branch is enriched with detailed and coherent semantic guidance, thereby enhancing visual quality and consistency. During training, the high-resolution image (*e.g.*, 4096 × 4096) is first down-sampled to the base resolution (1024 × 1024), then encoded

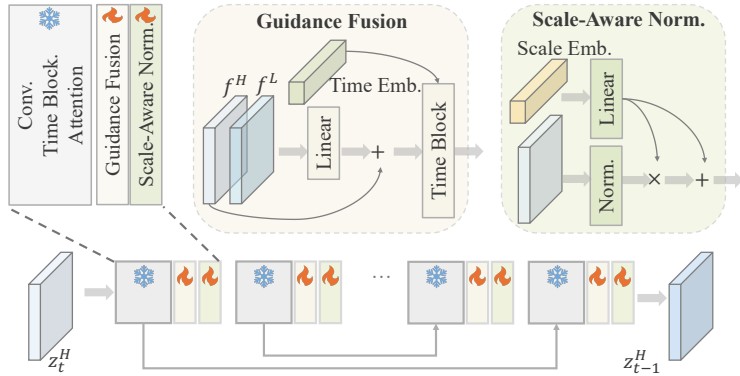

Figure 5: Architecture details of generative diffusion model.

to a latent $\mathbf{z}_0^L$ ($24 \times 24$) and corrupted with Gaussian noise as

$$q(\mathbf{z}_t^L|\mathbf{z}_0^L) := \mathcal{N}(\mathbf{z}_t^L; \sqrt{\overline{\alpha}_t}\mathbf{z}_0^L, (1 - \overline{\alpha}_t)\mathbf{I}), \tag{1}$$

where $\alpha_t := 1 - \beta_t$, $\sqrt{\overline{\alpha}_t} := \prod_{s=0}^{t} \alpha_s$, and $\beta_t$ is the pre-defined variance schedule for the diffusion process. We then feed $\mathbf{z}_t^L$ to the denoising network and obtain multi-level features after the attention blocks, denoted as the guidance features $\mathbf{g}$.

**Continuous upsampling.** Note that the guidance features $\mathbf{g}$ are at the base resolution ($24 \times 24$), while the HR features vary in size. To enhance our network's ability to utilize the guidance, we employ implicit neural representations [33, 5], which allow us to upsample the guidance features to arbitrary resolutions. This approach also mitigates noise disturbance in the guidance features, ensuring effective utilization of their semantic content. As shown in Figure 4, we initially perform dimensionality reduction on the LR guidance tokens via linear operartions for improved efficiency and concatenate them with a set of learnable tokens. These tokens undergo multiple self-attention layers, integrating information from the guidance features. Subsequently, the updated learnable tokens are processed through multiple linear layers to generate the implicit function weights. By inputting target position values into the implicit function, we obtain guidance features $\mathbf{g}'$ that matches the resolution of the HR features.

### 3.2 High-Resolution Latent Generation

The high-resolution latent generation is also conducted in the compact space (*i.e.*, $96 \times 96$ latent for a $4096 \times 4096$ image with a ratio of 1:42), significantly enhancing computational efficiency. Additionally, the high-resolution branch shares most of its parameters with the low-resolution branch, resulting in only a minimal increase in additional parameters. In detail, to incorporate LR guidance, we integrate several fusion modules. Furthermore, we implement resolution-aware normalization layers to adapt our model to varying resolutions.

**Guidance fusion.** After obtaining the guidance feature $\mathbf{g}'$, we fuse it with the HR feature $\mathbf{f}$ as follows:

$$\mathbf{f}' = \text{Linear}(\text{Concat}(\mathbf{f}, \mathbf{g}')) + \mathbf{f}. \tag{2}$$

The fused HR feature $\mathbf{f}'$ is further modulated by the time embedding $\mathbf{e}_t$ to determine the extent of LR guidance influence on the current synthesis step:

$$\mathbf{f}'' = \text{Norm}(\mathbf{f}') \odot \text{Linear}_1(\mathbf{e}_t) + \text{Linear}_2(\mathbf{e}_t) + \mathbf{f}'. \tag{3}$$

With such semantic guidance, our model gains an early understanding of the overall semantic structure, allowing it to fuse text information accordingly and generate finer details beyond the LR guidance, as illustrated in Figure 8.

**Scale-aware normalization.** As illustrated in Figure 2, changes in feature resolution result in corresponding variations in model representations. Normalization layers trained at a base resolution struggle to adapt to higher resolutions, such as $4096 \times 4096$. To address this challenge, we propose resolution-aware normalization layers to enhance model adaptability. Specifically, we derive the scale embedding $\mathbf{e}_s$ by calculating $\log_{N^H} N^L$, where $N^H$ denotes the number of pixels in the HR

features (*e.g.*, $96 \times 96$) and $N^L$ corresponds to the base resolution ($24 \times 24$). This embedding is then subjected to a multi-dimensional sinusoidal transformation, akin to the transform process used for time embedding. Finally, we modulate the HR feature $\mathbf{f}$ as follows:

$$\mathbf{f}' = \text{Norm}(\mathbf{f}) \odot \text{Linear}_1(\mathbf{e}_s) + \text{Linear}_2(\mathbf{e}_s) + \mathbf{f} \, . \tag{4}$$

The training objective of the generation process is defined as:

$$L := \mathbb{E}_{t, \mathbf{x}_0, \epsilon \sim \mathcal{N}(0,1)}[||\epsilon_{\theta, \theta'}(\mathbf{z}_t, s, t, \mathbf{g}) - \epsilon||_2] \, , \tag{5}$$

where $s$ and $\mathbf{g}$ denote scale and LR guidance, respectively. The parameters $\theta$ of the main generation network are fixed, while newly added parameters $\theta'$ including INR, guidance fusion, and scale-aware normalization are trainable.

## 4 Experiments

### 4.1 Implementation Details

We train models on 1M images of varying resolutions and aspect ratios, ranging from 1024 to 4608, sourced from LAION-Aesthetics [44], SAM [24], and self-collected high-quality dataset. The training is conducted on 8 A100 GPUs with a batch size of 64. Using model weight initialization from $1024 \times 1024$ StableCascade [36], our model requires only 15,000 iterations to achieve high-quality results. We employ the AdamW optimizer [30] with a learning rate of 0.0001. During training, we use continuous timesteps in $[0, 1]$ as [36], while LR guidance is consistently corrupted with noise at timestep $t = 0.05$. During inference, the generative model uses 20 sampling steps, and the diffusion decoding model uses 10 steps. We adopt DDIM [45] with a classifier-free guidance [19] weight of 4 for latent generation and 1.1 for diffusion decoding. Inference time is evaluated with a batch size of 1.

### 4.2 Comparison to State-of-the-Art Methods

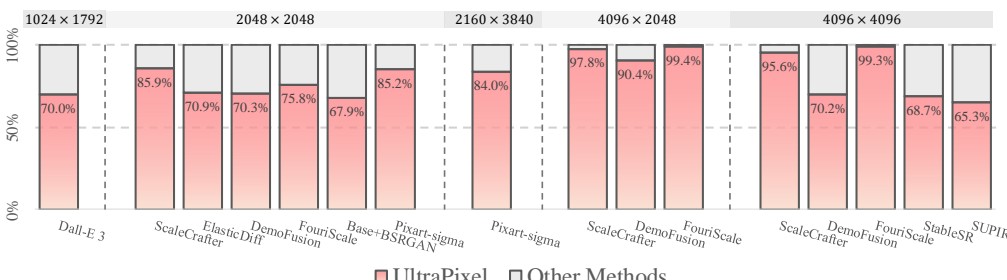

Figure 6: Win rate of our UltraPixel against competing methods in terms of PickScore [25].

**Compared methods**. We compare our method with competitive high-resolution image generation methods, categorized into training-free methods (ElasticDiffusion [14], ScaleCrafter [15], Fouriscale [21], Demofusion [12]) and training-based methods (Pixart-$\sigma$ [3], DALL·E 3 [34], and Midjourney V6 [32]). For models that can only generate $1024 \times 1024$ images, we use a representative image super-resolution method [51] for upsampling. We comprehensively evaluate the performance of our model at resolutions of $1024 \times 1792$, $2048 \times 2048$, $2160 \times 3840$, $4096 \times 2048$, and $4096 \times 4096$. For a fair comparison, we use the official implementations and parameter settings for all methods. Considering the slow inference time (tens of minutes to generate an ultra-high-resolution image) and the heavy computation of training-free methods, we compute all metrics using 1K images.

**Benchmark and evaluation**. We collect 1,000 high-quality images with resolutions ranging from 1024 to 4096 for evaluation. We focus primarily on the perceptual-oriented PickScore [25], which is trained on a large-scale user preference dataset to determine which image is better given an image pair with a text prompt, showing impressive alignment with human preference. Although FID [16] and Inception Score [43] (IS) may not fully assess the quality of generated images [25, 3], we report these metrics following common practice. It is important to note that both FID and IS are calculated on down-sampled images with a resolution of $299 \times 299$, making them unsuitable for evaluating

Table 1: Quantitative comparison with other methods. Our UltraPixel achieves state-of-the-art performance on all metrics across different resolutions.

| Resolution(H × W) | Method | $FID_P \downarrow$ | FID↓ | $IS_P \uparrow$ | IS↑ | CLIP↑ | Latency(sec.) ↓ |
|---|---|---|---|---|---|---|---|
| 1024 × 1792 | DALL·E 3 | 88.44 | 86.16 | 16.43 | 18.30 | 29.66 | - |
| | Ours | **60.5** | **63.53** | **17.84** | **26.89** | **35.34** | 8 |
| 2048 × 2048 | ScaleCrafter [15] | 64.75 | 73.79 | **15.41** | 22.53 | 31.79 | 45 |
| | ElasticDiffusion [14] | 77.19 | 65.37 | 11.12 | 21.97 | 32.95 | 295 |
| | DemoFusion [12] | 54.86 | 63.97 | 13.38 | 28.07 | 32.98 | 97 |
| | FouriScale [21] | 68.79 | 86.71 | 7.70 | 18.08 | 30.70 | 74 |
| | Base + BSRGAN [51] | 48.52 | 64.00 | 13.67 | 29.87 | 33.53 | 11+6 |
| | Pixart-Σ [3] | 54.35 | 63.96 | 14.87 | 27.13 | 31.18 | 57 |
| | Ours | **44.74** | **62.50** | 14.95 | **30.52** | **35.43** | **15** |
| 2160 × 3840 | Pixart-Σ [3] | 49.86 | 63.87 | 10.89 | 25.35 | 30.86 | 111 |
| | Ours | **46.06** | **62.41** | **11.91** | **25.65** | **34.98** | **31** |
| 4096 × 2048 | ScaleCrafter [15] | 101.58 | 120.71 | 9.04 | 12.15 | 23.71 | 190 |
| | DemoFusion [12] | 51.16 | 75.28 | 10.81 | 21.83 | 29.95 | 325 |
| | FouriScale [21] | 128.03 | 137.16 | 3.82 | 10.41 | 21.98 | 197 |
| | Ours | **42.60** | **64.69** | **11.76** | **25.36** | **34.59** | **33** |
| 4096 × 4096 | ScaleCrafter [15] | 74.02 | 98.11 | 9.07 | 14.53 | 31.79 | 580 |
| | DemoFusion [12] | 47.40 | **61.11** | 9.99 | 26.40 | 33.14 | 728 |
| | FouriScale [21] | 72.23 | 105.12 | 8.12 | 14.81 | 27.73 | 573 |
| | StableSR [48] | 48.18 | 65.27 | 9.25 | 27.55 | 32.49 | 728 |
| | SUPIR [50] | 46.98 | 64.13 | 9.83 | 26.16 | 31.28 | 682 |
| | Ours | **44.59** | 62.12 | **10.27** | **27.69** | **35.18** | **78** |

high-resolution image quality. Therefore, we adopt FID-patch and IS-patch for a more reasonable measure. Finally, we evaluate image-text consistency using the CLIP score [7].

**Quantitative Comparison**. As mentioned, PickScore aligns closely with human perception, so we use it as our primary metric. Figure 6 shows the win rate of our UltraPixel compared to other methods. Our approach consistently delivers superior results across all resolutions. Notably, UltraPixel is preferred in $85.2\%$ and $84.0\%$ of cases compared to the training-based Pixart-Σ [3], despite Pixart-Σ using separate parameters for different resolutions and training on 33M images, whereas our model uses the same parameters for varying resolutions and is trained on just 1M images. UltraPixel also shows competitive performance compared to advanced T2I commercial product DALL·E 3 [34], yielding a win rate of $70.0\%$. Continuous LR guidance enables our resolution-aware model to focus on detail synthesis, resulting in higher visual quality. Furthermore, as shown in Table 1, our method performs competitively on FID, IS, and CLIP scores across different resolutions. Training-free HR generation methods [12, 15, 21, 14] struggle to produce high-quality $4096 \times 2048$ images, showing limited generalization ability. Our UltraPixel also excels in inference efficiency, generating a $2160 \times 3840$ image in 31 seconds, which is nearly $3.6\times$ faster than Pixart-Σ (111 seconds). Compared to training-free methods that take tens of minutes to generate a $4096 \times 4096$ image, our model is significantly more efficient, being $9.3\times$ faster than DemoFusion [12]. These results highlight the effectiveness of our method in generating ultra-high-resolution images with excellent efficiency.

**Qualitative comparison**. Figure 7 illustrates a visual comparison between our UltraPixel and other high-resolution image synthesis methods at various resolutions. Training-free methods like ScaleCrafter [15] and FouriScale [21] often produce visually unpleasant structures and large areas of irregular textures, significantly degrading visual quality. DemoFusion [12] suffers from severe small object repetition due to its patch-by-patch generation approach. Compared to Pixart-Σ [3], our method excels in generating superior semantic coherence and fine-grained details. For instance, in the $2160 \times 3840$ resolution case, our generated camel and human faces exhibit richer details. Despite using a single model to generate images at different resolutions, our method consistently produces visually pleasing and semantically coherent results. Besides, as illustrated in Figure B.5, B.6, and B.7 of appendix, our method produces images of quality comparable to those generated by DALL·E 3 and Midjourney V6.

### 4.3 Ablation Study

In this section, for computational efficiency, we train all models with 5K iterations. Unless otherwise stated, the results are reported at a resolution of $2560 \times 2560$.

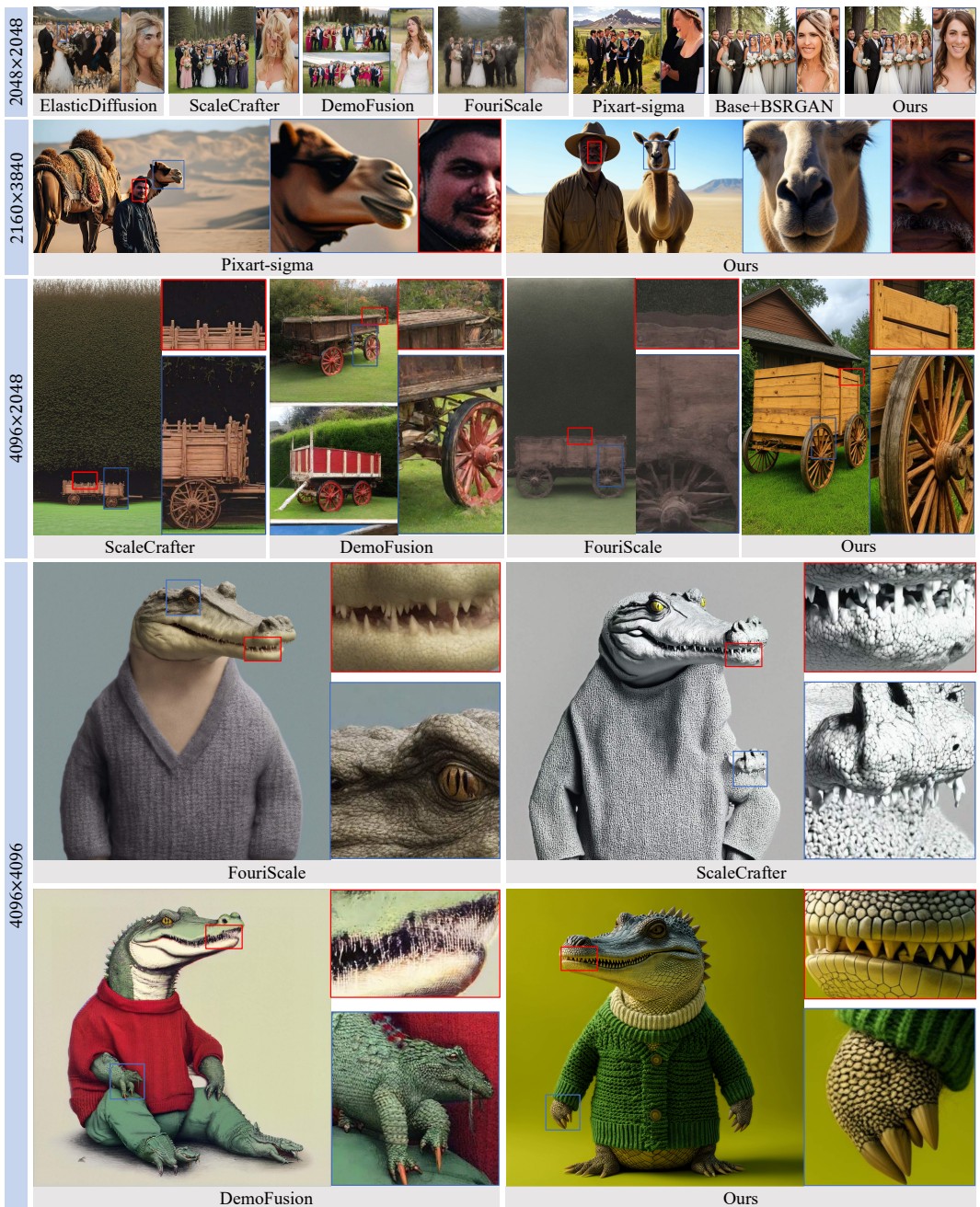

Figure 7: Visual Comparison of our UltraPixel and other methods. Our method produces images of ultra-high resolution with enhanced details and superior structures. More visual examples are provided in the appendix.

**LR guidance.** Figure 8 visually demonstrates the effectiveness of LR guidance. The synthesized HR result without LR guidance exhibits noticeable artifacts, with a messy overall structure and darker color tone. In contrast, the HR image generated with LR guidance is of higher quality, for instance, the characters "*accepted*" on the sweater and the details of the fluffy head are more distinct. Visualization of attention maps reveals that the HR image generation process with LR guidance shows clearer structures earlier. This indicates that LR guidance provides strong semantic priors for HR generation, allowing the model to focus more on detail refinement while maintaining better semantic coherence. Additionally, Figure 9 compares our method to the post-processing super-resolution strategy, demonstrating that UltraPixel can generate more visually pleasing details.

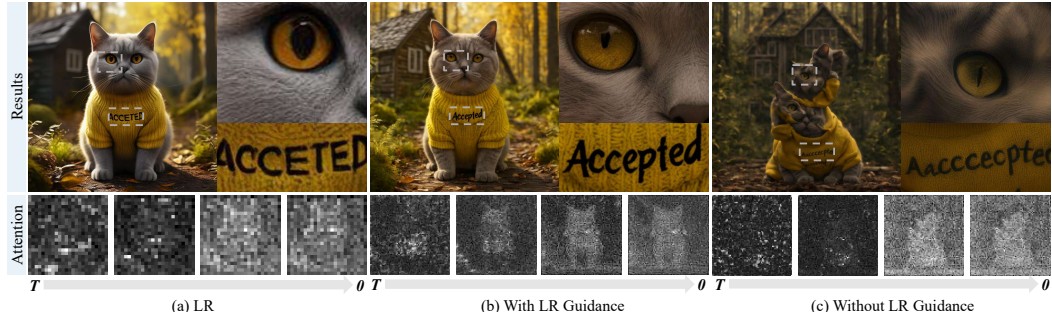

|   |   |   |
|---|---|---|
| **T** ──────────→ **0** | **T** ──────────→ **0** | **T** ──────────→ **0** |
| (a) LR | (b) With LR Guidance | (c) Without LR Guidance |

Figure 8: Ablation study on LR guidance. Leveraging the semantic guidance from LR features allows the HR generation process to focus on detail refinement, improving visual quality. Text prompt: *In the forest, a British shorthair cute cat wearing a yellow sweater with "Accepted" written on it. A small cottage in the background, high quality, photorealistic, 4k.*

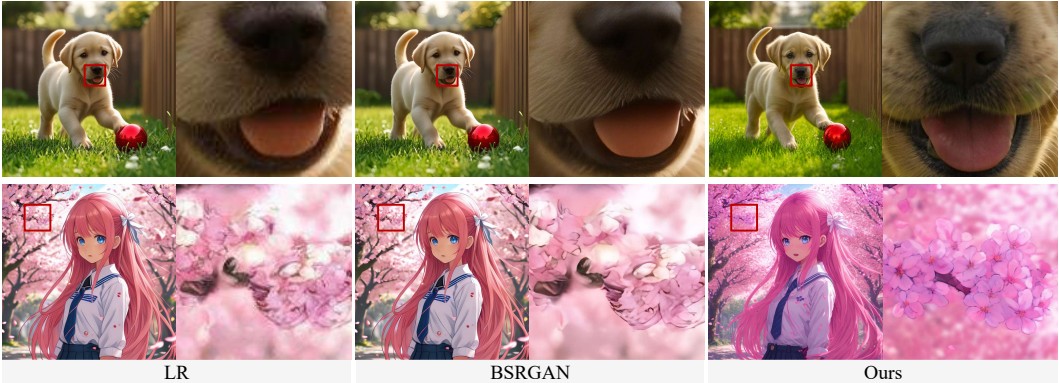

|   |   |   |
|---|---|---|
| LR | BSRGAN | Ours |

Figure 9: Visual comparison with super-resolution method BSRGAN [51] at resolution of $4096 \times 4096$. Super-resolution has limited ability to refine the details of the low-resolution image, while our method is capable of generating attractive details.

**Timesteps of LR guidance extraction.** We analyze the effect of timesteps used to extract LR guidance in Table 2 and Figure 10. We consider three cases: $t = t^H$, where LR guidance is synchronized with the HR timesteps; $t = 0.5$, representing a fixed guidance at the middle timestep; and $t = 0.05$, near the end. The results show that $t = t^H$ produces a poor CLIP score. This can be attributed to the necessity of providing semantic structure guidance early on, but the LR guidance is too noisy at this stage to be useful. Similarly, $t = 0.5$ also results in noisy LR guidance, as seen in Figure 8. Conversely, $t = 0.05$ provides the best performance since features in the later stage of generation exhibit much clearer structural information. With semantics-rich guidance, HR image generation can produce coherent structures and fine-grained details, yielding higher scores in Table 2.

**Implicit neural representation (INR).** To incorporate multi-resolution capability into our model, we adopt an INR design to continuously provide informative semantic guidance. In Table 3, we compare continuous INR upsampling (dubbed "INR") with directly upsampling LR guidance using bilinear interpolation followed by convolutions (denoted as "BI + Conv"). The results show that INR yields better semantic alignment and image quality, as it provides consistent guidance of LR features across varying resolutions. Figure 11 further illustrates that directly upsampling LR guidance introduces significant noise into the HR generation process, resulting in degraded visual quality.

**Scale-aware normalization.** As illustrated in Figure 2, features across different resolutions vary significantly. To generate higher-quality results, we propose scale-aware normalization (SAN). Table 3 compares the performance of models with ("INR + SAN") and without ("INR") this design. When scaling the resolution from $2560 \times 2560$ to $4096 \times 4096$, the CLIP score gap noticeably enlarges, indicating better textual alignment with SAN. Additionally, the Inception Score shows significant improvement when adopting SAN, validating the effectiveness of our design.

Table 2: Ablation study on timesteps of LR guidance extraction.

| | $t = t^H$ | $t = 0.5$ | $t = 0.05$ |
|---|---|---|---|
| CLIP↑ | 31.14 | 32.75 | **33.09** |
| IS↑ | 25.37 | 28.15 | **29.14** |

Table 3: Ablation on INR and SAN.

| | | BI + Conv | INR | INR + SAN |
|---|---|---|---|---|
| $2560^2$ | CLIP↑ | 32.41 | 32.72 | **33.09** |
| | IS↑ | 26.81 | 27.62 | **29.14** |
| $4096^2$ | CLIP↑ | 31.90 | 31.93 | **32.87** |
| | IS↑ | 22.22 | 25.22 | **27.15** |

Table 4: Ablation on the number of trainable parameters.

| | Base | LoRA | Ours-512 | Ours-1024 |
|---|---|---|---|---|
| Param.(M) | 0 | 106 | 65 | 101 |
| CLIP↑ | 30.39 | 31.20 | 32.78 | **33.09** |
| IS↑ | 20.89 | 22.73 | 27.43 | **29.14** |

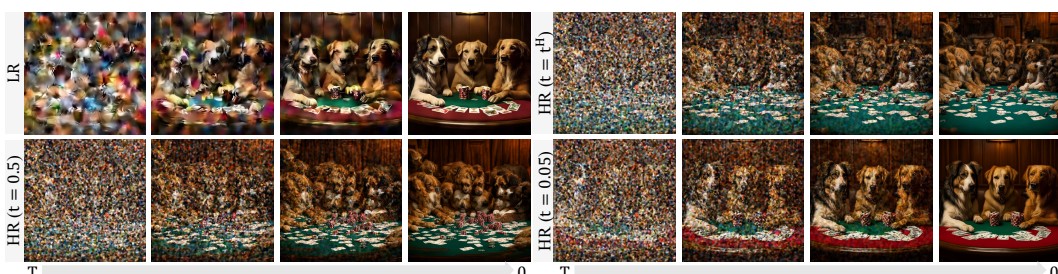

Figure 10: Visual comparisons of guidance across different timesteps. "LR" depicts the low-resolution generation process, whereas the other "HR" cases illustrate the high-resolution process under varying guidance. When employing synchronized ($t = t^H$) or middle timestep ($t = 0.5$) guidance, the structure information provided is messy, while $t = 0.05$ offers semantics-rich and clear directives.

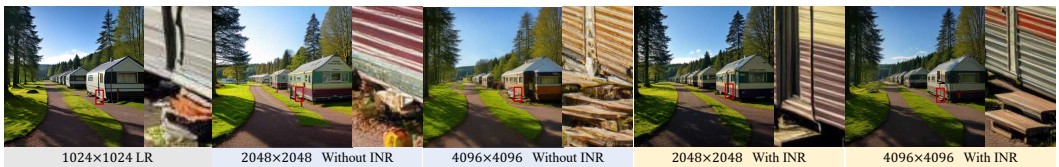

Figure 11: Illustration of Implicit neural representation (INR) to provide consistent guidance.

**Number of trainable parameters.** Our model benefits from high training efficiency, partly because we use a limited number of trainable parameters based on StableCascade [36]. Table 4 illustrates the impact of the number of trainable parameters. Since most new parameters are in the INR module, we can reduce the channel dimension of LR features from 2048 to a lower number. We explore models with LR dimensions of 512 and 1024 and also include a LoRA [20] version with a rank of 48. Compared to the "LoRA" model, "Ours-512" produces better results with fewer parameters. Increasing the channel number from 512 to 1024 ("Ours-1024") achieves higher visual quality and better text-image alignment. To balance efficiency and performance, we choose 1024 as the default.

# 5 Conclusion

We present UltraPixel, an efficient framework for generating high-quality images at varying resolutions. Utilizing an extremely compact latent space, we introduce low-resolution (LR) guidance to simplify the complexity of semantic planning and detail synthesis. Specifically, semantics-rich LR features provide structural guidance for high-resolution image generation. To enable our model to handle varying resolutions, we learn an implicit function to consistently upsample LR features and insert scale-aware normalization layers to adapt feature distribution. UltraPixel efficiently generates stunning, ultra-high-resolution images of varying sizes, elevating image synthesis to new heights.

# 6 Broader Impacts and Limitation

Despite the advancements in UltraPixel, the limited quantity and quality of training datasets constrain the realism and quality of our generated images, especially in complex scenes. This issue underscores the ongoing challenges in achieving true photorealism, and we are committed to further exploring this area in future research.

# 7 Acknowledgments

This work is supported by the Guangzhou-HKUST(GZ) Joint Funding Program (No. 2023A03J0671), the Guangzhou Municipal Science and Technology Project (Grant No. 2024A04J4230), Guangdong Provincial Key Lab of Integrated Communication, Sensing and Computation for Ubiquitous Internet of Things(No.2023B1212010007), and the National Natural Science Foundation of China (Project No. 61902275).

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

## Appendix

In Section A, we first demonstrate that the latent space of StableCascade [36] can accommodate images with various resolutions and compare the reconstruction quality with SDXL [37]. Subsequently, in Section B, we provide additional visual comparisons with the super-resolution method, cutting-edge high-resolution generation techniques, and leading closed-source T2I products. We also present more high-resolution results of our method in Section C. Next, we illustrate how our model can be customized for controllable generation and personalization in Section D. Finally, we include text prompts for the images generated, presented in both the main document and the appendix in Section E.

## A  Latent Space of StableCascade

As illustrated in Figure A.1, StableCascade [36] achieves a high compression ratio of 42:1 while capably reconstructing images of varying sizes with promising quality. Although there is some loss of detail, this is considered acceptable given the significant efficiency gains in both training and inference that the high compression ratio facilitates. In contrast, as shown in Table A.1, SDXL [37] has a lower compression ratio 8:1 and obtains higher PSNR scores, indicating superior fidelity between the reconstructed images and the original high-resolution inputs. Considering the trade-off between efficiency and accuracy, we emphasize the value of StableCascade's compact representation and its suitability for ultra-high-resolution generation applications.

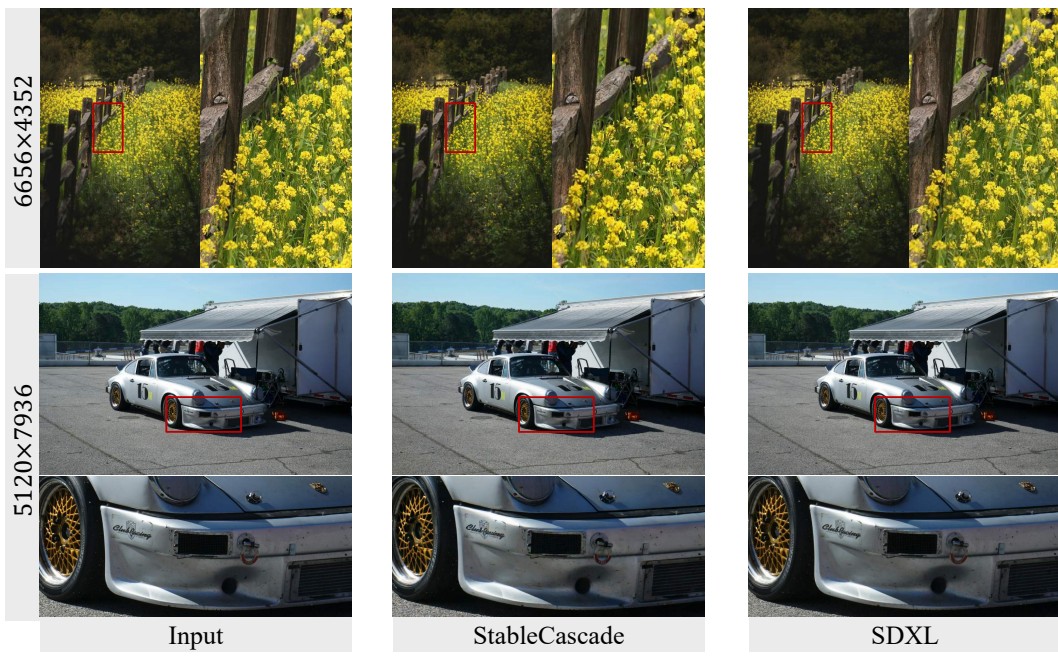

Figure A.1: Visual comparison of reconstruction quality between VAEs of StableCascade [36] and SDXL [37] on high-resolution images.

Table A.1: Quantatitive comparison of reconstruction quality and complexity between StableCascade [36] and SDXL [37]

| StableCascade | | | SDXL | | |
|---|---|---|---|---|---|
| PSNR↑ | Compress Ratio | # of Params (M) | PSNR↑ | Compress Ratio | # of Params (M) |
| 30.87dB | 42 : 1 | 1520 | 33.08dB | 8:1 | 80 |

# B   Additional Comparison Results

**Comparison with an SR method.** A common method to obtain high-resolution images involves initially generating a low-resolution image and then upsampling it with an off-the-shelf super-resolution (SR) model. In Figure B.2 and B.3, we compare the results produced by our UltraPixel method and the advanced super-resolution techniques, BSRGAN [51], StableSR [47] and SUPIR [50]. It is evident that the SR method often fails to introduce adequate details; although the resolution increases, the image quality does not improve proportionately. In contrast, our UltraPixel method excels by incorporating an abundance of intricate details, significantly enhancing the visual quality of the images.

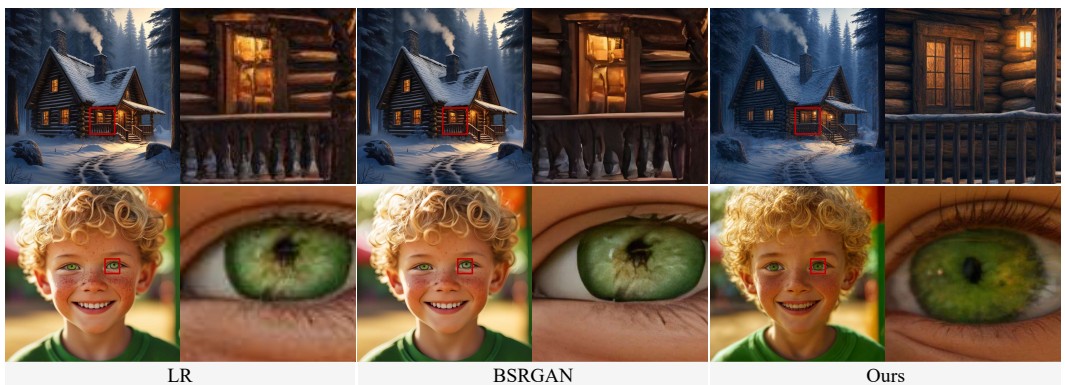

Figure B.2: Visual comparison with BSRGAN [51] at $4096 \times 4096$ resolution.

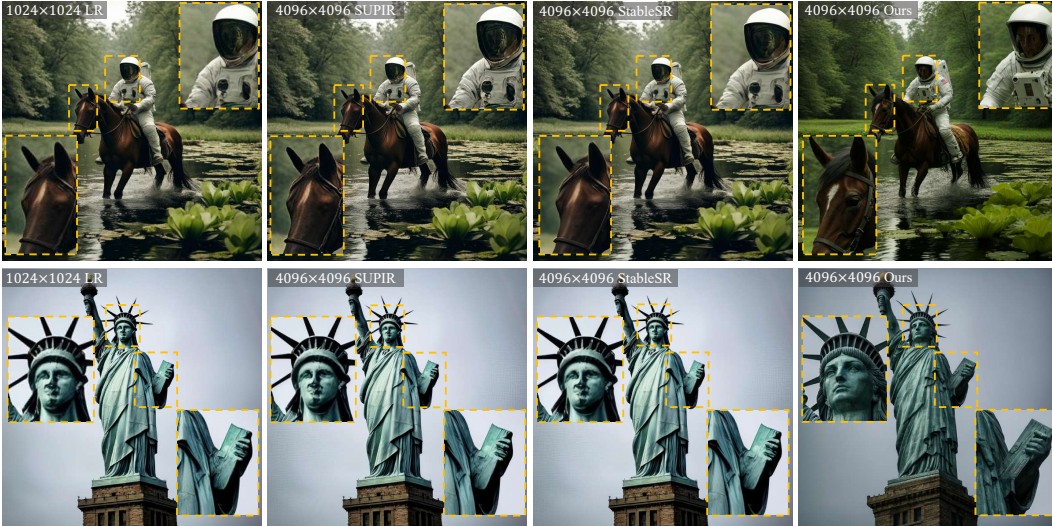

Figure B.3: Visual comparison with generative diffusion-based super-resolution method StableSR [47] and SUPIR [50] at $4096 \times 4096$ resolution.

**Comparison with high-resolution image generation methods.** We present additional visual comparisons with state-of-the-art high-resolution image generation methods in Figure B.4. The results generated by our UltraPixel method consistently outperform others across various resolutions, highlighting its superior capability.

**Comparison with closed-source T2I products.** We offer further visual comparisons between our UltraPixel and closed-source commercial text-to-image (T2I) products: DALL·E 3 [34] in Figure B.5 and Midjourney V6 [32] in Figures B.6 and B.7. Our method showcases the ability to generate high-quality images that are on par with these leading commercial products.

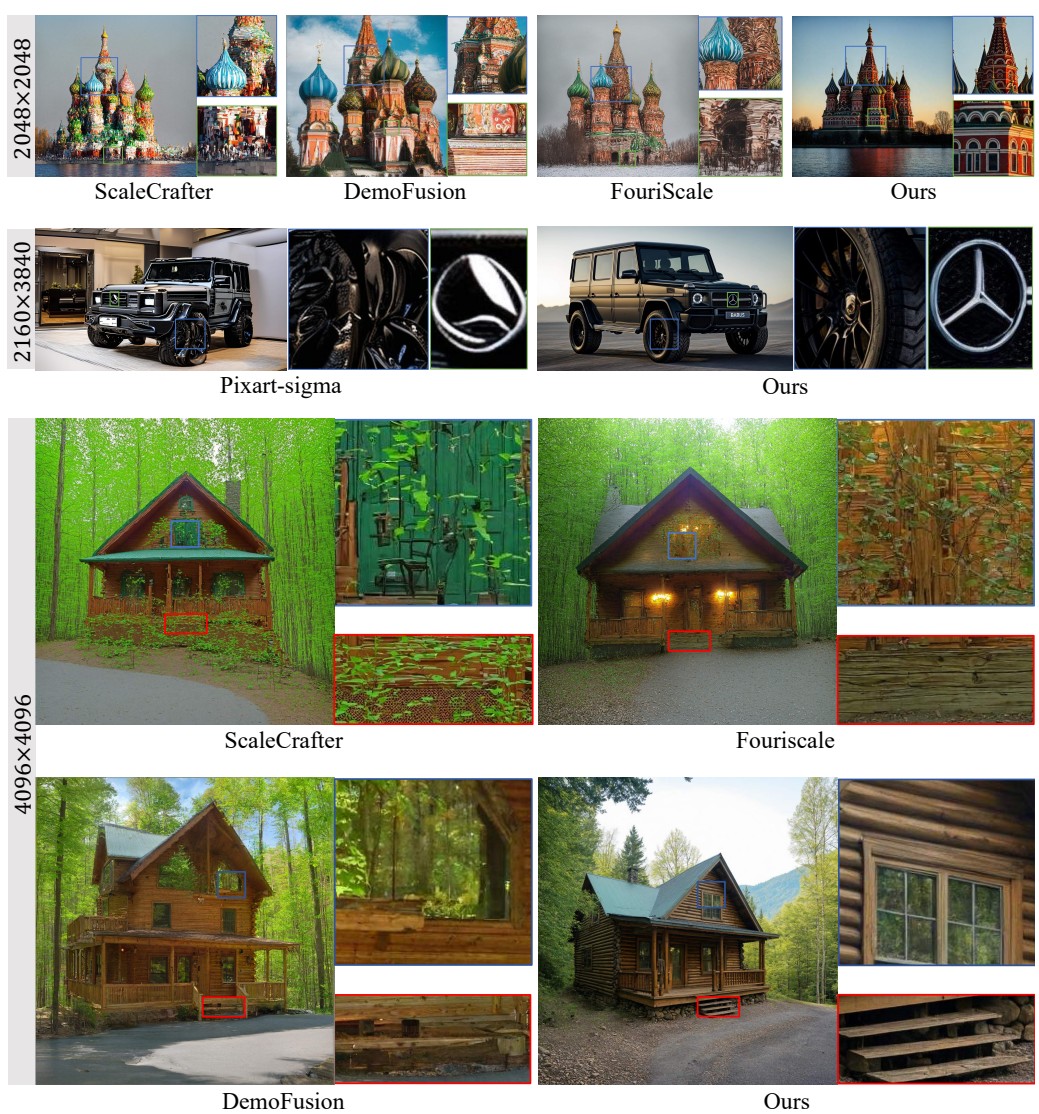

Figure B.4: Visual comparison with high-resolution image generation methods.

## C  Additional Visual Results

We present more visual results of UltraPixel in Figure C.8, C.9, C.10, C.11, C.12, C.13, C.14, C.15, C.16. Our method produces images of diverse resolutions with excellent quality, excelling in a range of scenarios from close-up portraits and imaginative content to photo-realistic scenes.

## D  Controllable High-Resolution Image Synthesis

**Spatial control.** We present high-resolution (HR) results controlled by edge maps. Notably, we do not train our models directly; rather, we utilize the officially released control weights from StableCascade [36]. These control features are integrated during the low-resolution (LR) guidance extraction process. The results are demonstrated in Figures D.17 and D.18. Currently, the maximum supported resolution is 3K. Further fine-tuning of the control weights will enable support for higher resolutions.

**Personalization.** Figure D.19 demonstrates high-resolution personalized results based on a user-provided instance. Specifically, we optimize the model parameters of the attention layers using

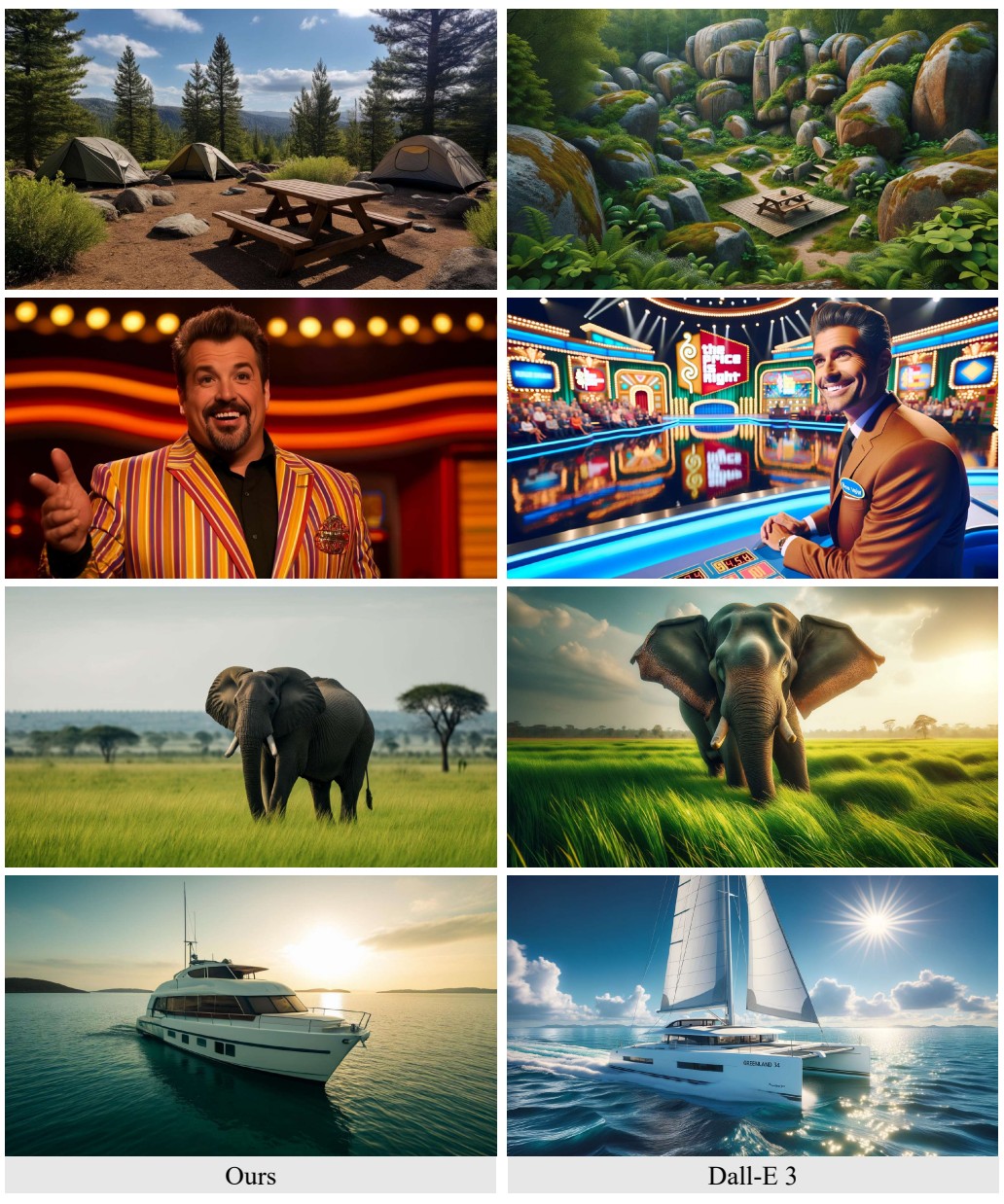

| Ours | Dall-E 3 |

Figure B.5: Visual comparison with Dall·E 3 [34] at $1024 \times 1792$ resolution.

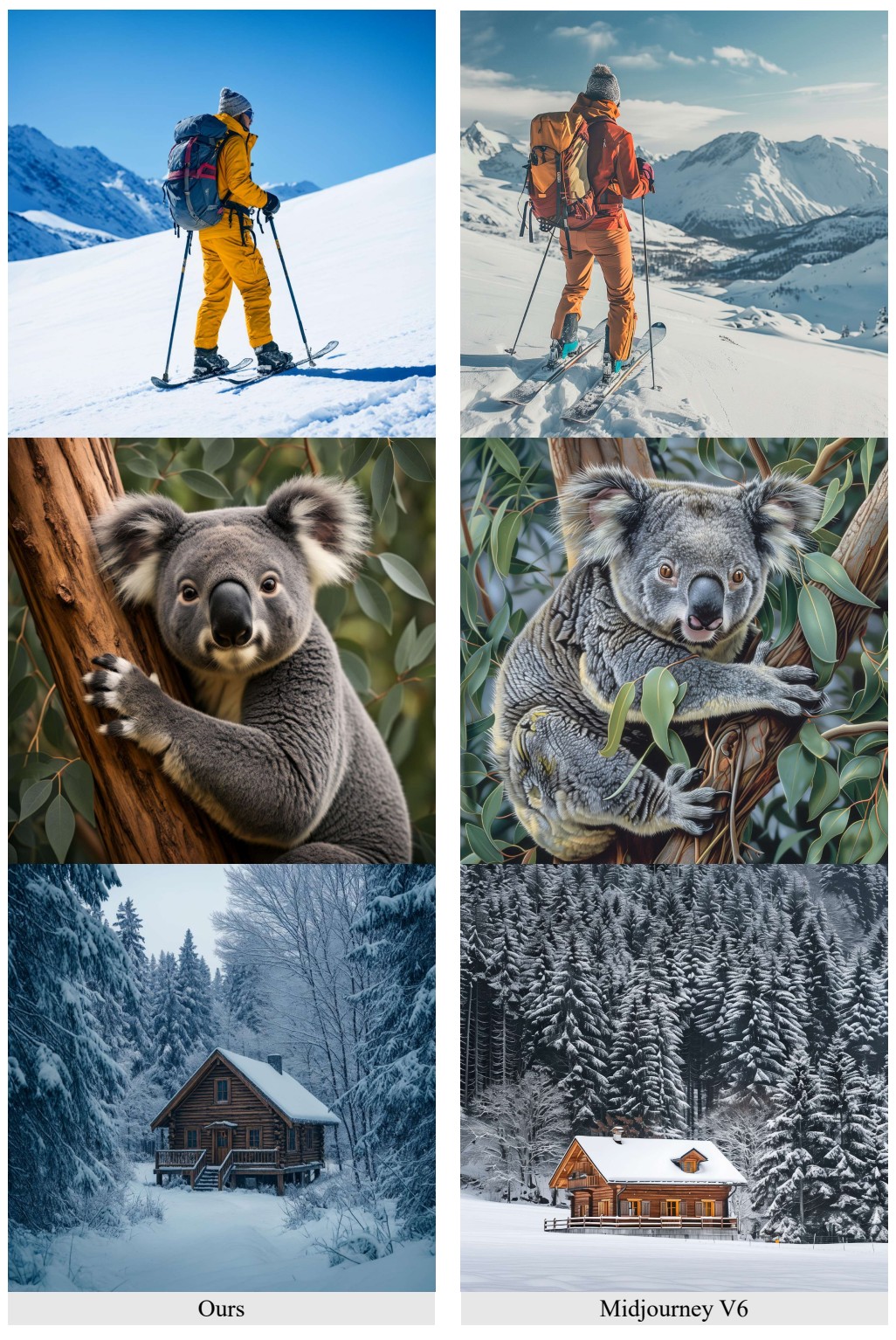

| Ours | Midjourney V6 |

Figure B.6: Visual comparison with Midjourney V6 [32] at $2048 \times 2048$ resolution.

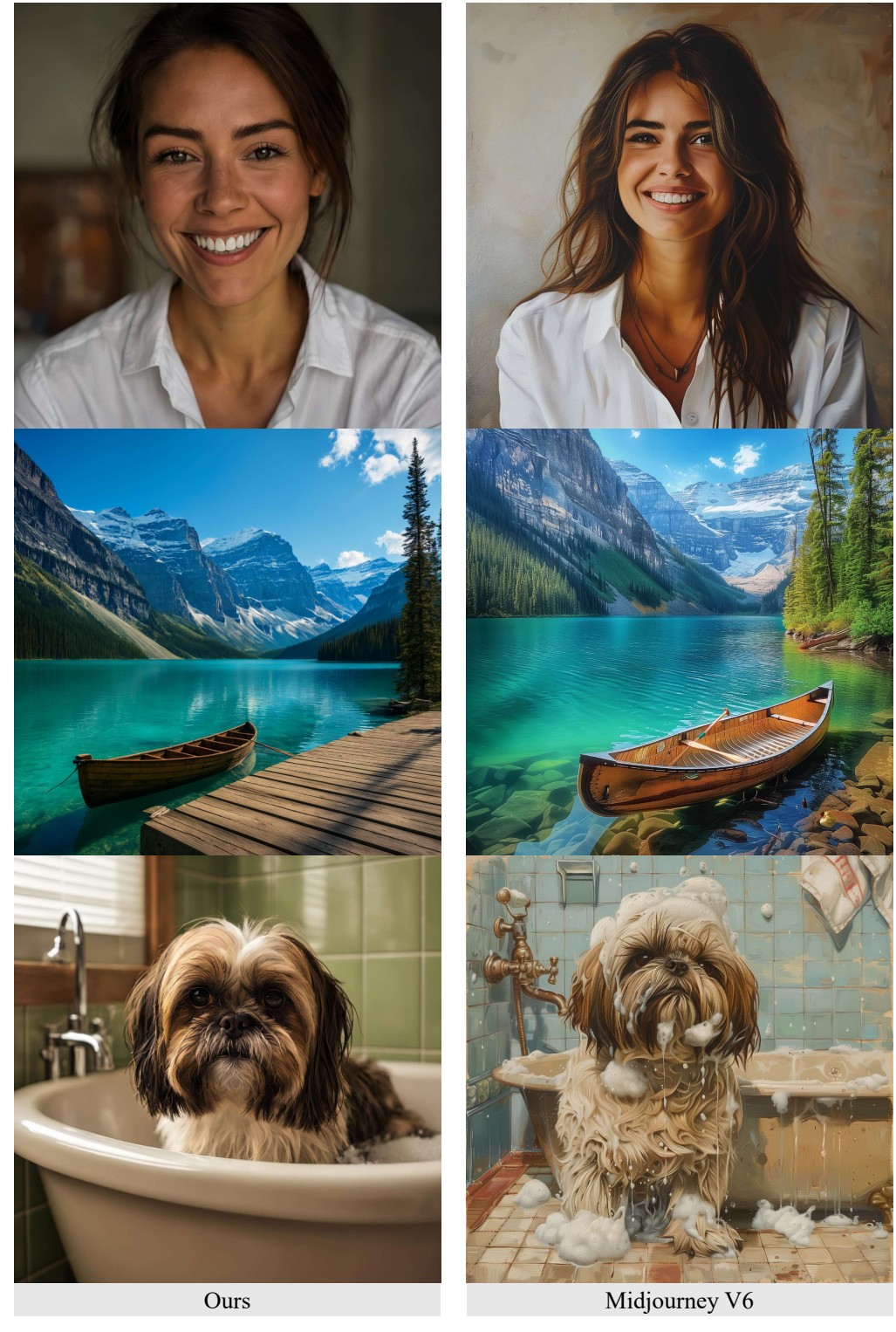

| Ours | Midjourney V6 |

Figure B.7: Visual comparison with Midjourney V6 [32] at $2048 \times 2048$ resolution.

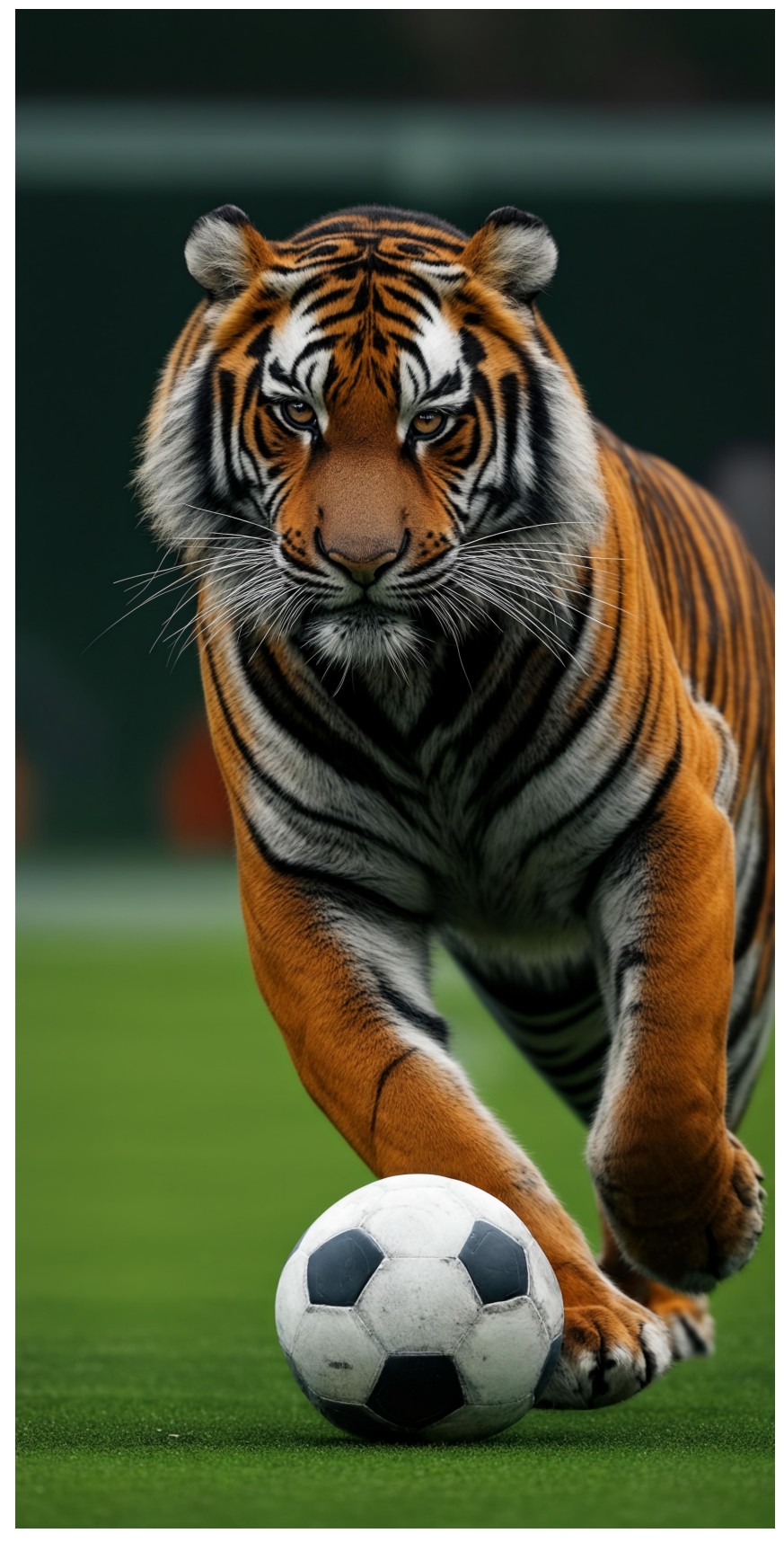

Figure C.8: Visual results of UltraPixel at $5120 \times 2560$ resolution.

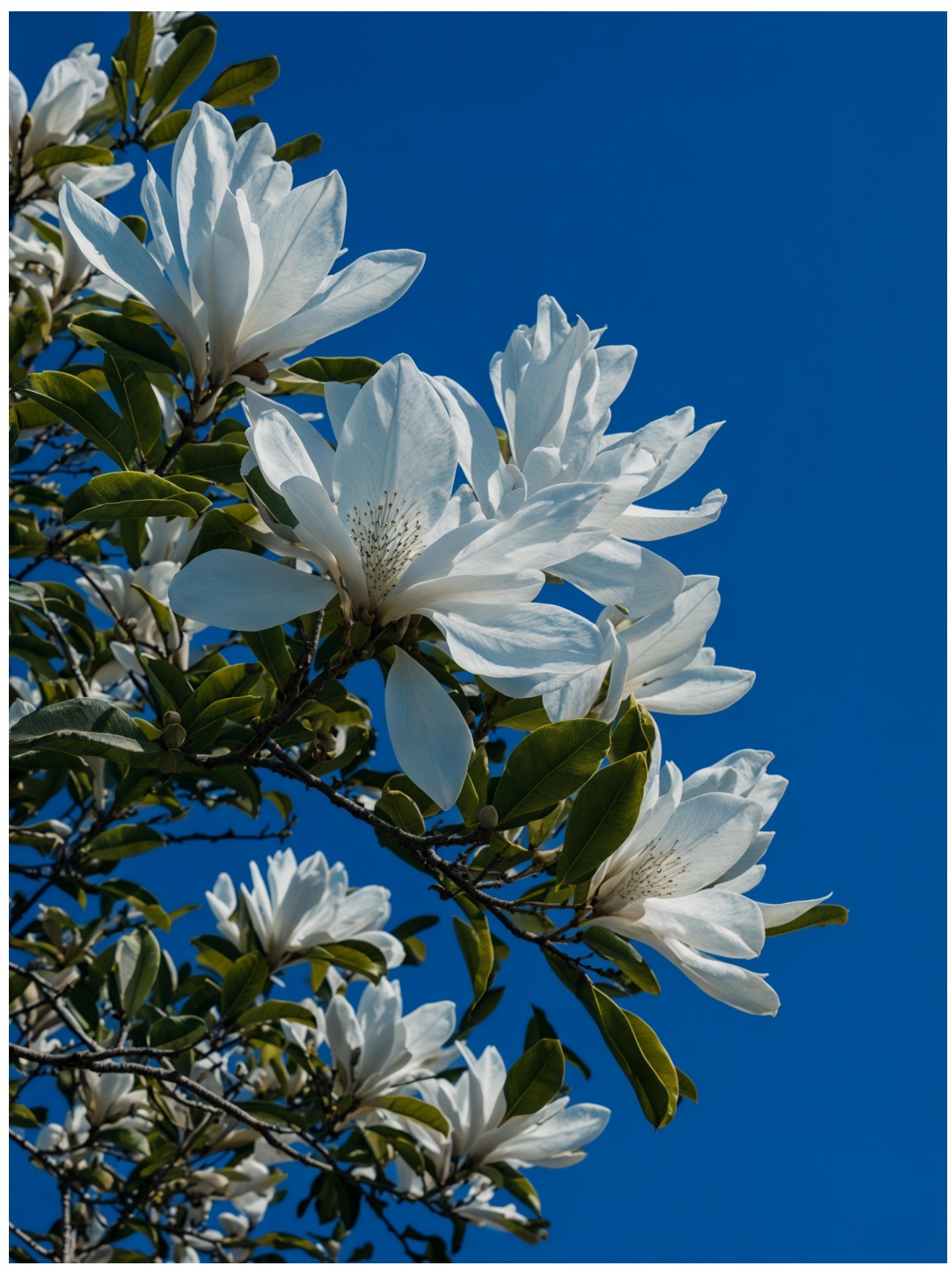

Figure C.9: Visual results of UltraPixel at $5120 \times 3840$ resolution.

LoRA [20] with a rank of 4. The training involves an initial phase at a base resolution for 5,000 iterations, followed by fine-tuning at a higher resolution for an additional 5,000 iterations. Figure D.19 showcases our method's capability to incorporate personalized techniques for achieving personalized high-resolution image generation.

## E Text Prompts

Text prompts are provided in Table E.2, E.3.

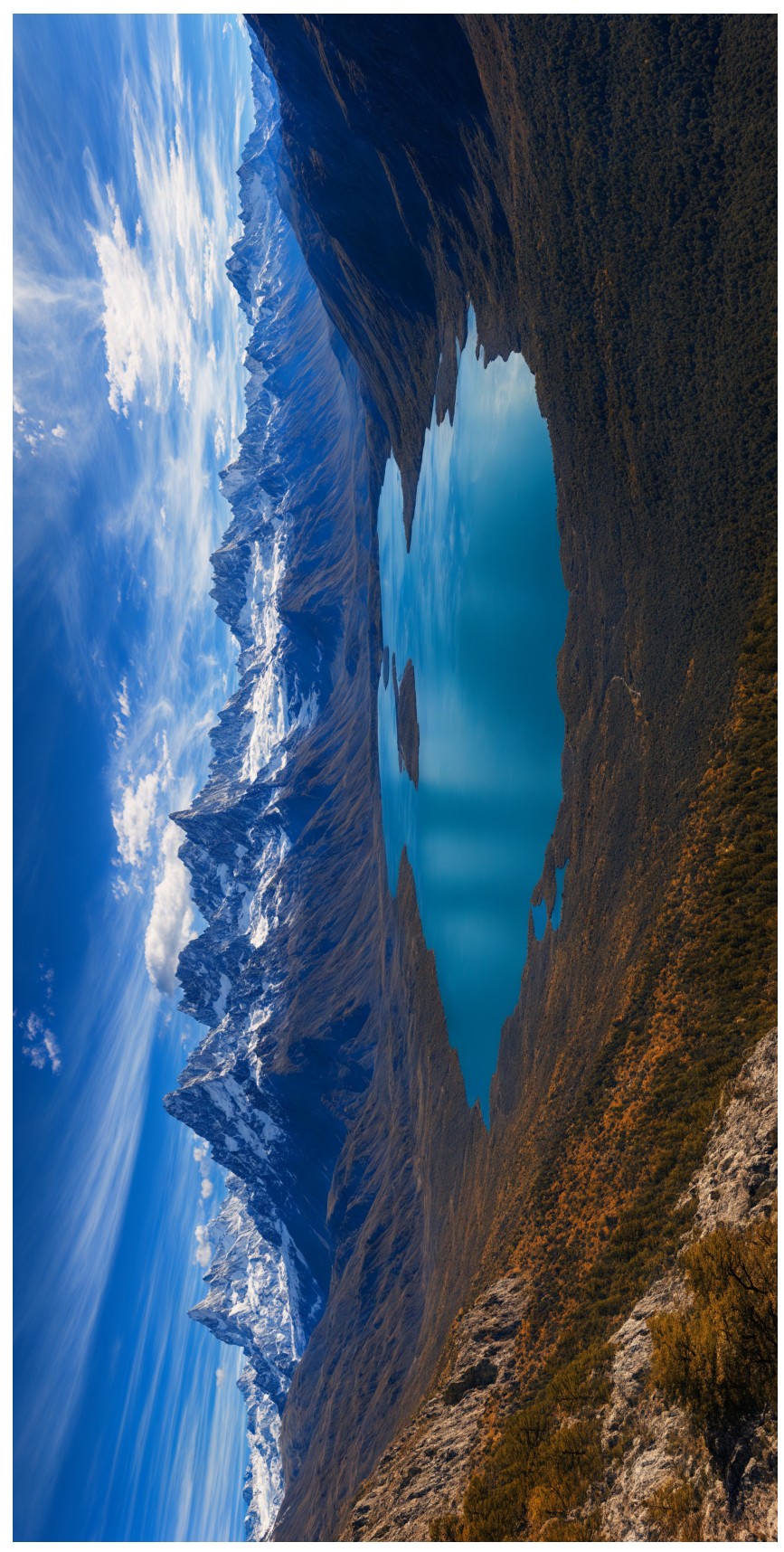

Figure C.10: Visual results of UltraPixel at $3072 \times 6144$ resolution.

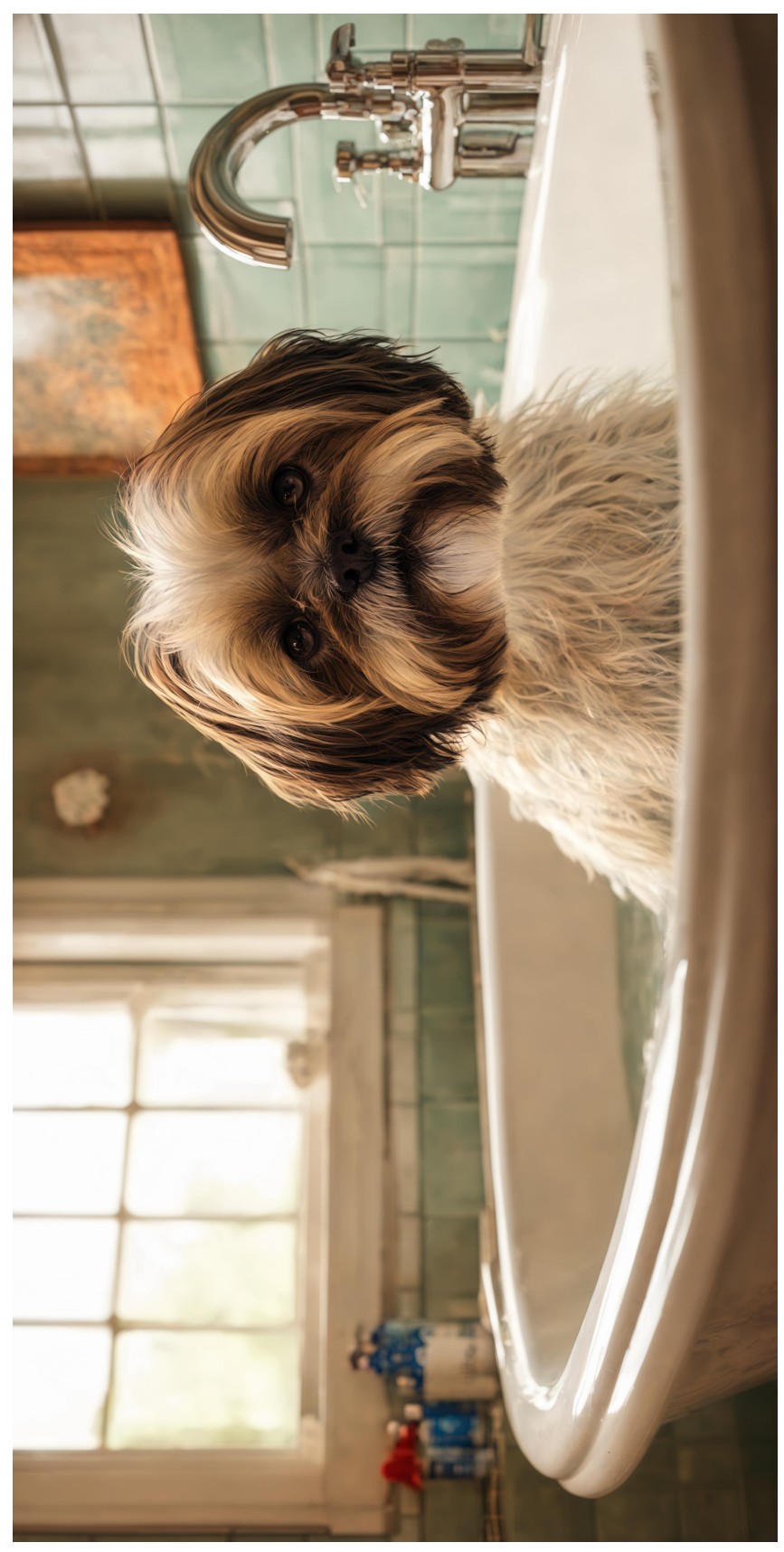

Figure C.11: Visual results of UltraPixel at $3072 \times 6144$ resolution.

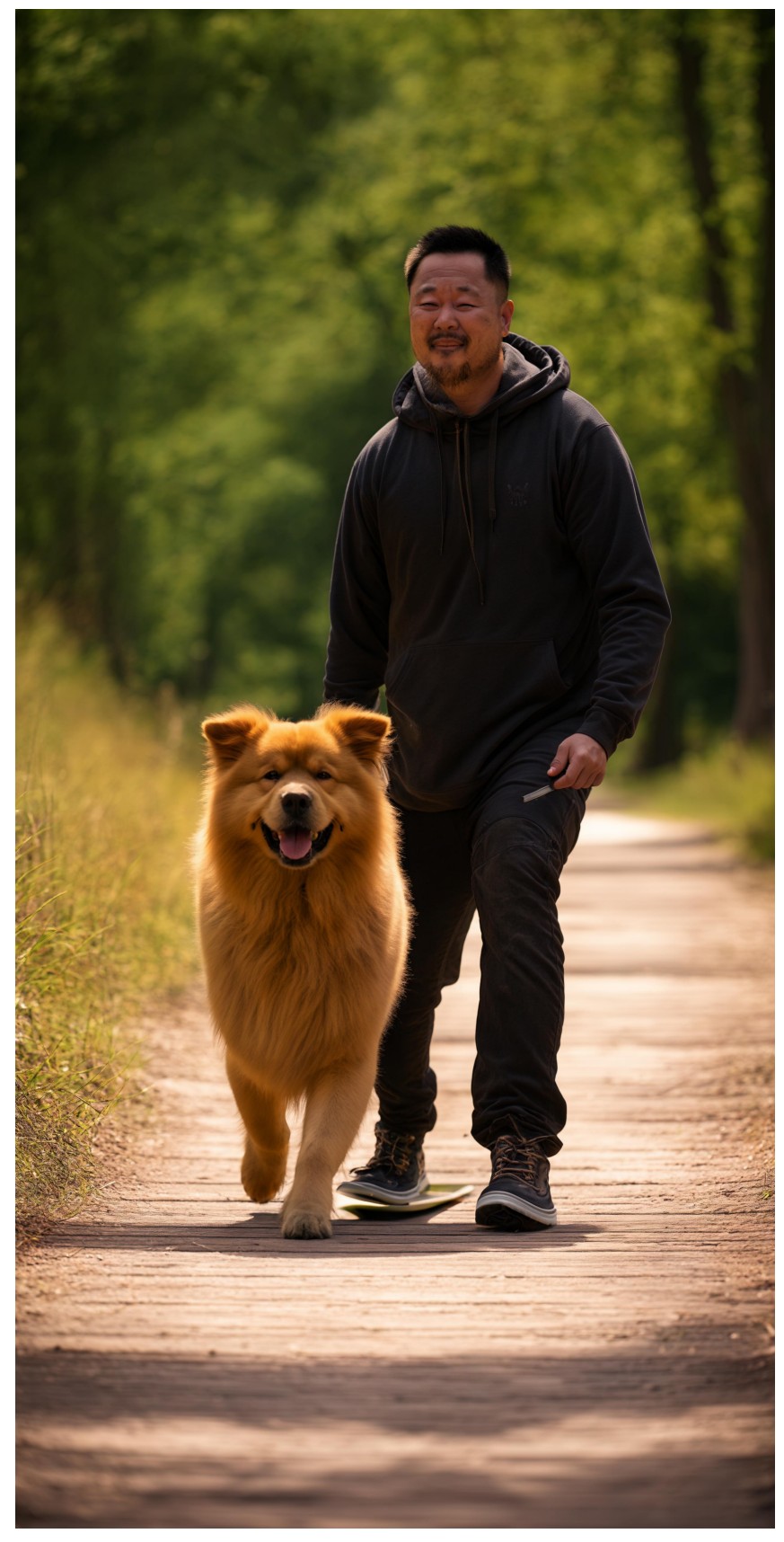

Figure C.12: Visual results of UltraPixel at $5120 \times 2560$ resolution.

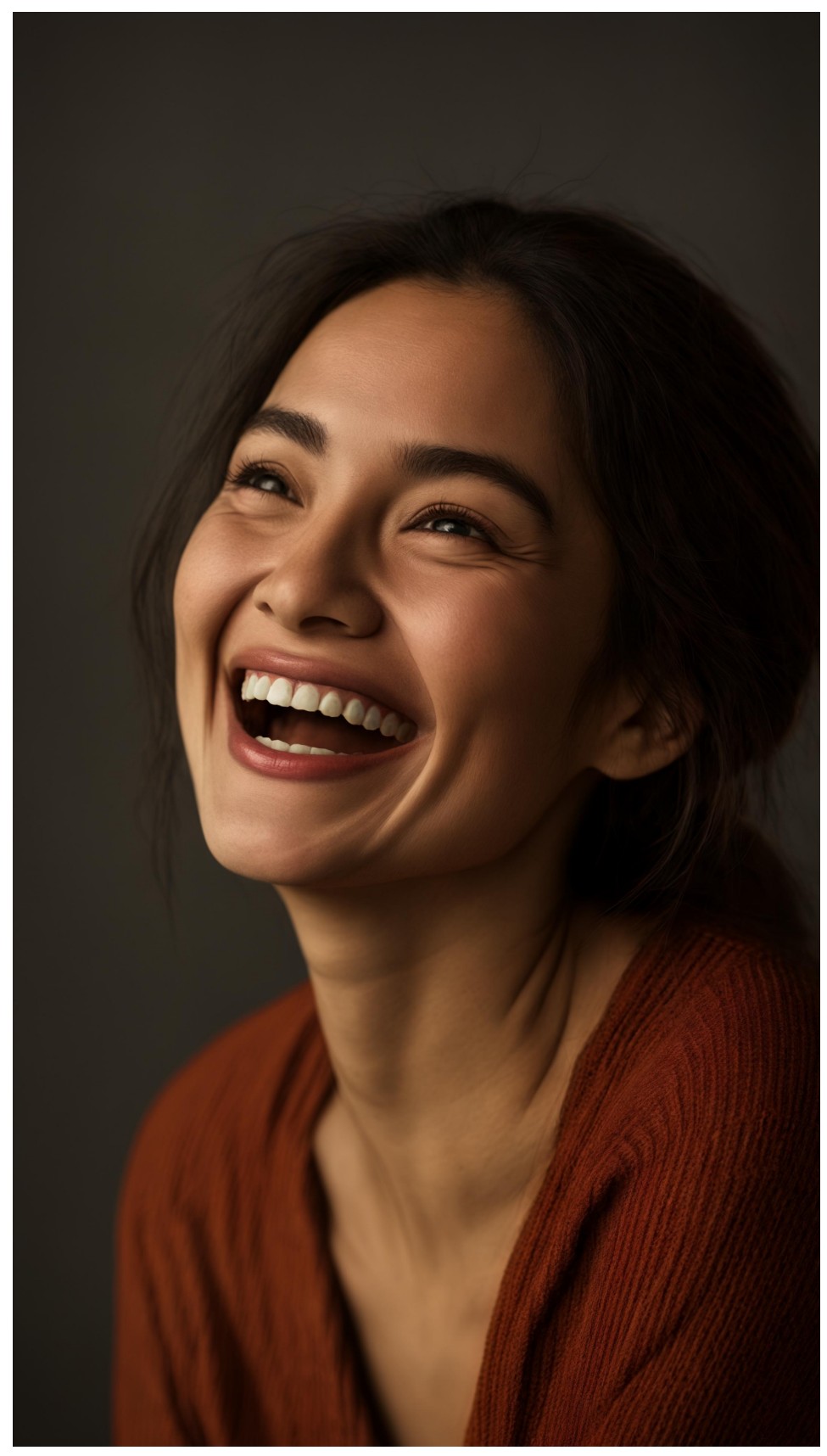

Figure C.13: Visual results of UltraPixel at $3840 \times 2160$ resolution.

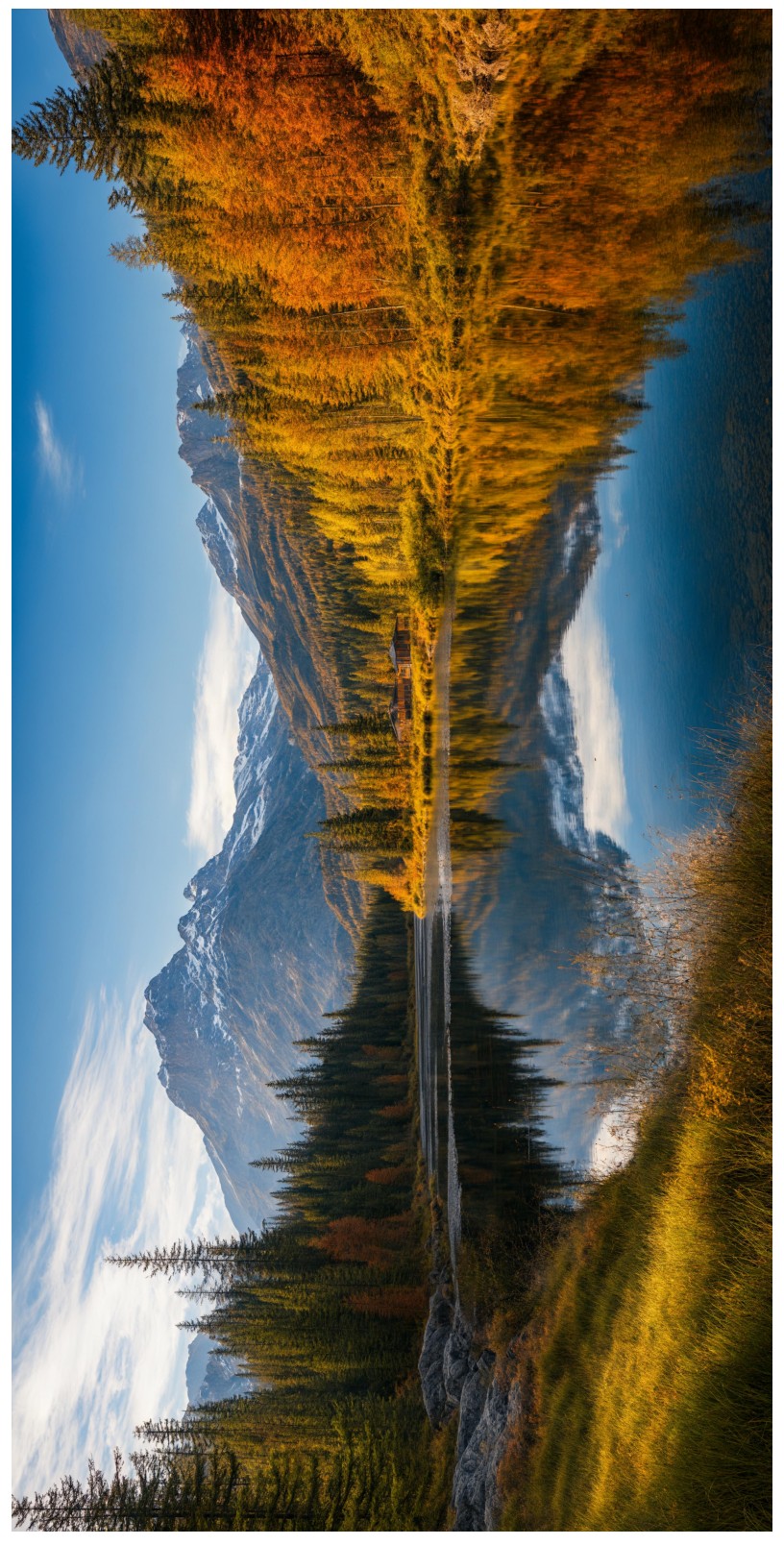

Figure C.14: Visual results of UltraPixel at $2880 \times 5760$ resolution.

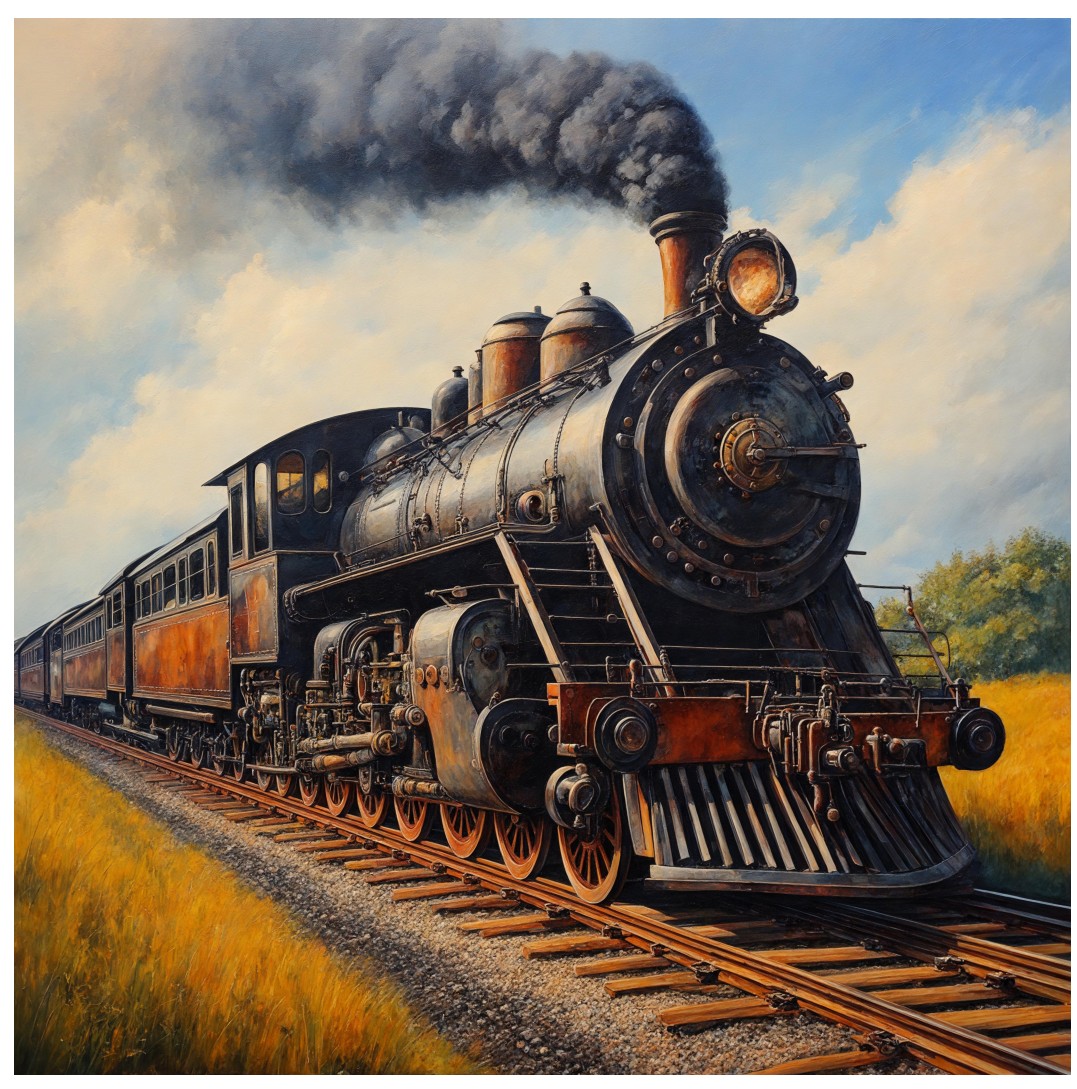

Figure C.15: Visual results of UltraPixel at $4096 \times 4096$ resolution.

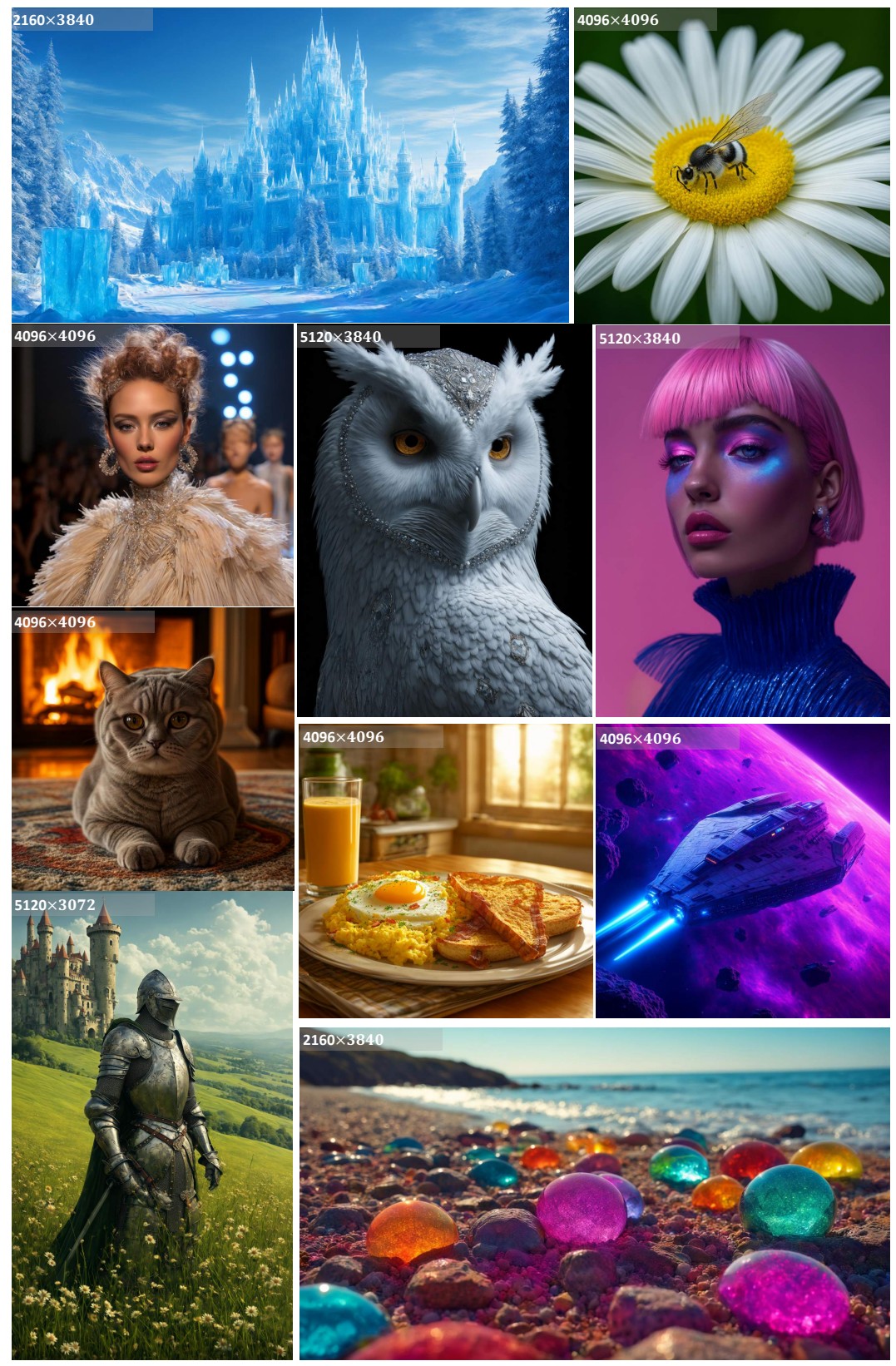

Figure C.16: Visual results of UltraPixel.

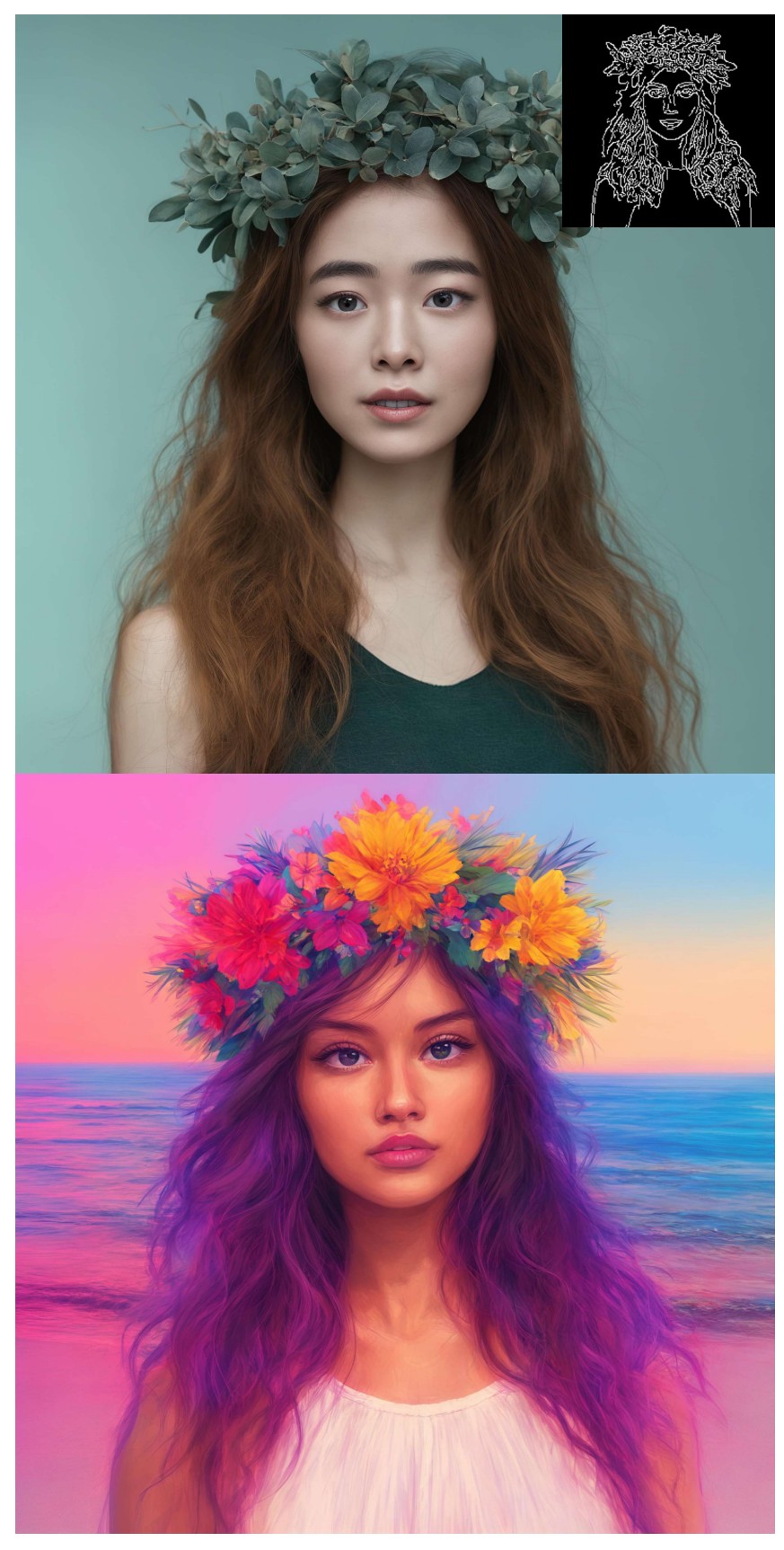

Figure D.17: Edge-controlled results of UltraPixel at $3072 \times 3072$ resolution.

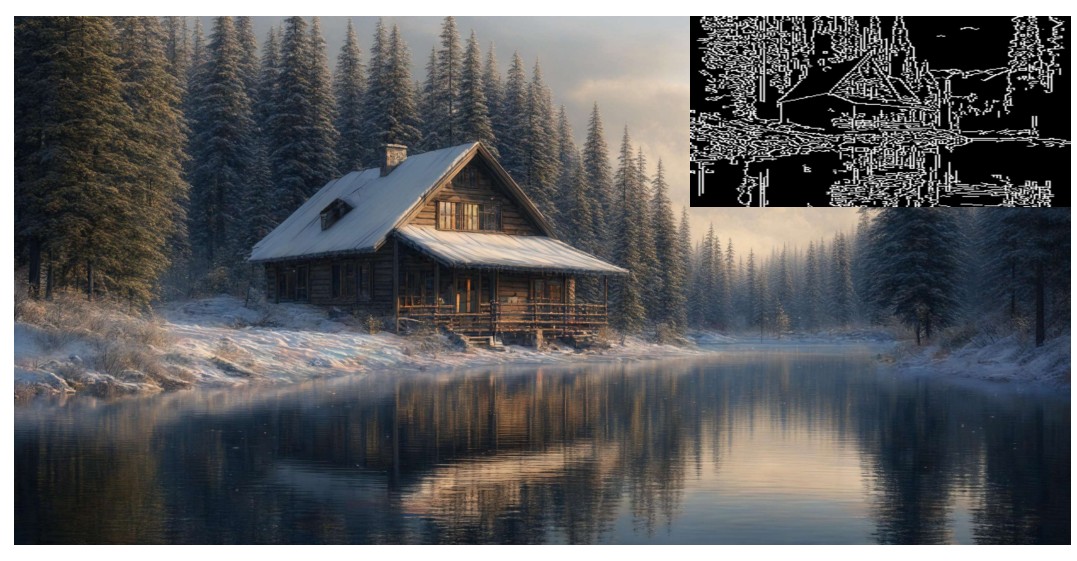

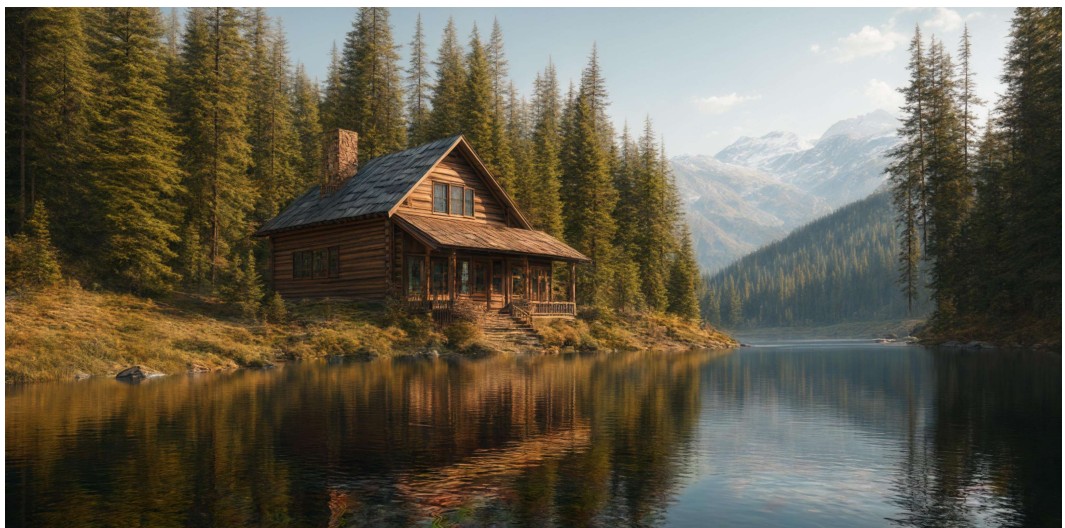

Figure D.18: Edge-controlled results of UltraPixel at $2160 \times 3840$ resolution.

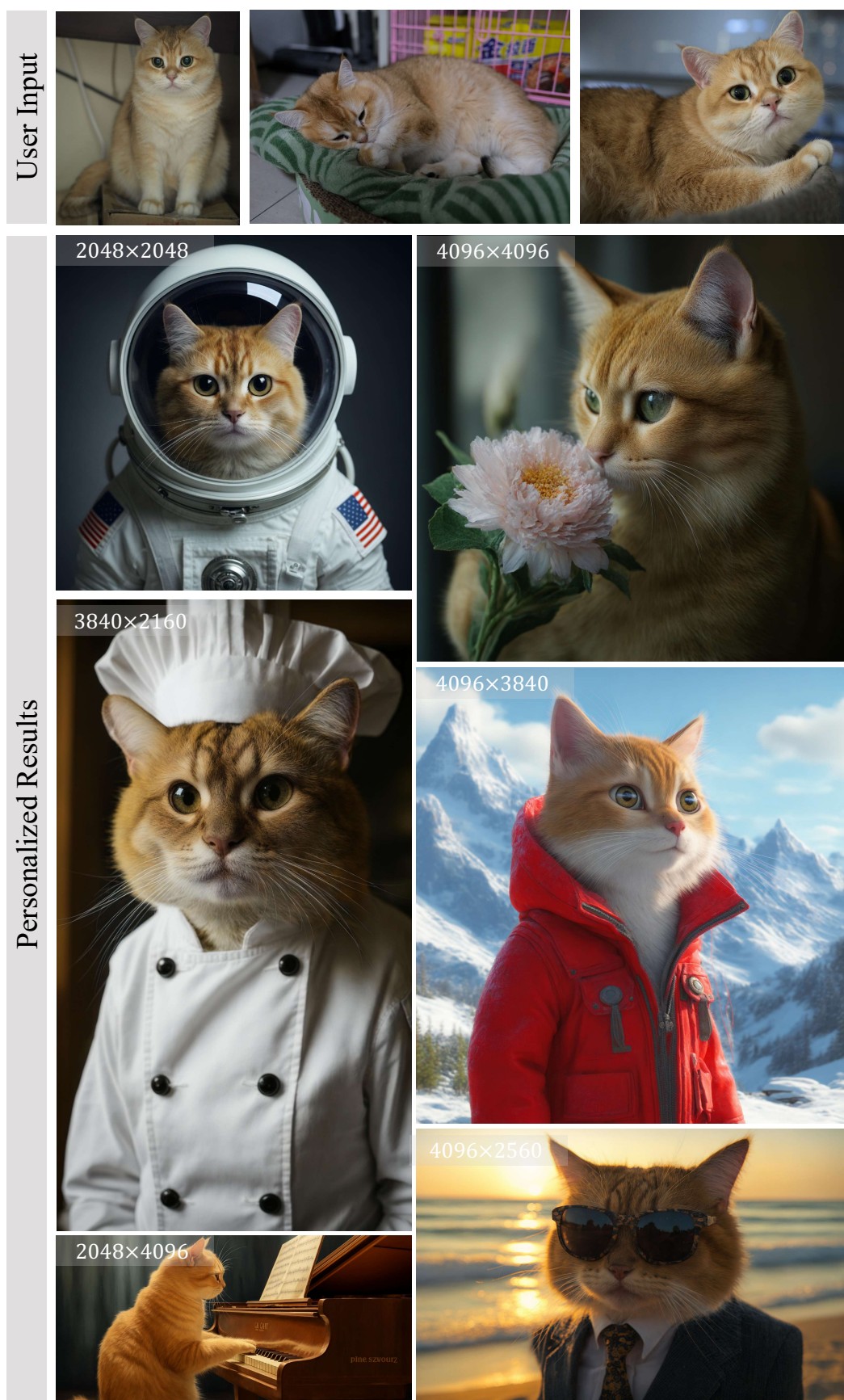

Figure D.19: Personalization results of UltraPixel.

Table E.2: Text prompts of figures in this manuscript. For each figure, prompts are provided in order from left to right, top to bottom. (to be continued)

| Figure | Text Prompt |
|---|---|
| Figure 1 | A close-up of a blooming peony, with layers of soft, pink petals, a delicate fragrance, and dewdrops glistening in the early morning light. |
| | A serene mountain landscape with towering snow-capped peaks, a crystal-clear blue lake reflecting the mountains, dense pine forests, and a vibrant orange sunrise illuminating the sky. |
| | An expressionist landscape with twisted trees, a turbulent sky, and a path leading to an unknown destination, painted in vivid, unsettling colors and expressive brushstrokes that convey a sense of anxiety and movement. |
| | A close-up portrait of a young woman with flawless skin, vibrant red lipstick, and wavy brown hair, wearing a vintage floral dress and standing in front of a blooming garden. |
| | The image features a snow-covered mountain range with a large, snow-covered mountain in the background. The mountain is surrounded by a forest of trees, and the sky is filled with clouds. The scene is set during the winter season, with snow covering the ground and the trees. |
| | A statue of a person holding a torch |
| | A pair of cuddly rabbits, one white with floppy ears and the other brown with a twitching nose, snuggling together in a cozy hutch filled with straw. |
| Figure 2 | Ford anglia van |
| Figure 7 | Idaho wedding party |
| | A man standing next to a camel |
| | A wooden wagon in a yard |
| | Crocodile in a sweater |
| Figure 9 | A vibrant anime scene of a young girl with long, flowing pink hair, big sparkling blue eyes, and a school uniform, standing under a cherry blossom tree with petals falling around her. The background shows a traditional Japanese school with cherry blossoms in full bloom. |
| | A playful Labrador retriever puppy with a shiny, golden coat, chasing a red ball in a spacious backyard, with green grass and a wooden fence. |
| Figure 10 | Dogs sitting around a poker table |
| Figure 11 | Blairgowrie Holiday Park, Blairgowrie, Perthshire \| Head Outside |
| Figure B.2 | A cozy, rustic log cabin nestled in a snow-covered forest, with smoke rising from the stone chimney, warm lights glowing from the windows, and a path of footprints leading to the front door. |
| | A striking close-up of a young boy with curly blonde hair and bright green eyes, his face sprinkled with freckles. He is smiling widely, showcasing a gap-toothed grin, and the background is a sunny, out-of-focus playground. |
| Figure B.5 | Campsite with picnic table surrounded by boulders and green plants. |
| | Joey Fatone Hosts The Price Is Right - Live Show At Bally's Las Vegas |
| | steam rises from a geyser in a mountainous area |
| | FOUNTAINE PAJOT Greenland 34 |
| Figure B.6 | A person with a backpack and skis in the snow |
| | A charming depiction of a koala resting in a eucalyptus tree, with the soft gray fur and the lush green leaves creating a peaceful scene. |
| | brown wooden house in the middle of snow covered trees |
| Figure B.7 | Smiling woman in white shirt |
| | A highly detailed, high-quality image of the Banff National Park in Canada. The turquoise waters of Lake Louise are surrounded by snow-capped mountains and dense pine forests. A wooden canoe is docked at the edge of the lake. The sky is a clear, bright blue, and the air is crisp and fresh. |
| | A highly detailed, high-quality image of a Shih Tzu receiving a bath in a home bathroom. The dog is standing in a tub, covered in suds, with a slightly wet and adorable look. The background includes bathroom fixtures, towels, and a clean, tiled floor. |
| Figure B.4 | SSt. Basil's Cathedral |
| | 2014 brabus b63s 700 6x6 mercedes benz g class hd pictures. Black Bedroom Furniture Sets. Home Design Ideas |
| | Ext for in ldg and sc gatlinburg cabin wahoo sale night cabins rentals of american homes tn log city |
| Figure C.8 | A tiger is playing football. |
| Figure C.9 | A detailed view of a blooming magnolia tree, with large, white flowers and dark green leaves, set against a clear blue sky. |
| Figure C.10 | A highly detailed, high-quality image of the Patagonia region in Argentina. Towering mountains with snow-covered peaks rise above pristine lakes and dense forests. Glaciers can be seen in the distance, reflecting the bright sunlight. The sky is a deep blue with scattered white clouds. |
| Figure C.11 | A highly detailed, high-quality image of a Shih Tzu receiving a bath in a home bathroom. The dog is standing in a tub, covered in suds, with a slightly wet and adorable look. The background includes bathroom fixtures, towels, and a clean, tiled floor. |
| Figure C.12 | Flowrider taking his chow chow for a walk. |
| Figure C.13 | A laughing woman |

Table E.3: Text prompts of figures in this manuscript. For each figure, prompts are provided in order from left to right, top to bottom. (continued)

| Figure | Text Prompt |
|---|---|
| Figure C.14 | Adventure Romance Trip Ideas tree outdoor sky grass reflection Nature wilderness tarn mountain mountainous landforms water leaf nature reserve Lake highland pond mount scenery loch national park fell landscape bank biome cloud wetland autumn hill tundra mountain range River valley larch meadow Forest surrounded lush hillside |
| Figure C.15 | A detailed and realistic painting of a vintage steam train, with intricate details in the machinery and a sense of motion and power, capturing the nostalgia and romance of the era, highly detailed, high quality |
| Figure C.16 | Ice Kingdom: A stunning ice kingdom with crystalline castles and frozen landscapes. The castles are made entirely of ice, with spires that sparkle in the sunlight. Snow-covered trees and icy pathways lead to the grand palace at the heart of the kingdom, where the ice queen resides. The sky above is a brilliant blue, with snowflakes gently falling. |
| | A detailed macro shot of a daisy, showcasing its white petals and bright yellow center, with tiny insects like bees and butterflies hovering nearby. |
| | A high-fashion runway show featuring models in avant-garde clothing, dramatic makeup, and elaborate hairstyles, with bright lights and a stylish, modern backdrop. |
| | Anthropomorphic profile of the white snow owl Crystal priestess, art deco painting, pretty and expressive eyes, ornate costume, mythical, ethereal, intricate, elaborate, hyperrealism, hyper detailed, 3D, 8K, Ultra Realistic, high octane, ultra resolution, amazing detail, perfection, In frame, photorealistic, cinematic lighting, visual clarity, shading, lumen reflections, super-resolution, gigapixel, color grading, retouch, enhanced, PBR, Blender, V-ray, procreate, zBrush. |
| | Art collection style and fashion shoot, in the style of made of glass, dark blue and light pink, paul rand, solarpunk, camille vivier, beth didonato hair, barbiecore, hyper-realistic |
| | A highly detailed, high-quality image of a Scottish Fold cat sitting on a bookshelf. The cat's distinctive folded ears and round face give it a unique appearance as it sits among books and decorative items. The background features a well-lit room with wooden shelves and a reading nook |
| | Traditional Breakfast: A hearty traditional breakfast with fluffy scrambled eggs, crispy bacon, golden hash browns, and buttered toast. The plate is garnished with fresh parsley and accompanied by a glass of freshly squeezed orange juice and a steaming cup of coffee. The background features a cozy kitchen setting with a morning sunbeam streaming through the window. |
| | Space Adventure: A thrilling scene of a spaceship navigating through an asteroid field. The ship is sleek and futuristic, with glowing thrusters and advanced weaponry. The asteroids are large and rugged, illuminated by the light of distant stars. In the background, a nebula in vibrant hues of blue and purple adds a touch of cosmic beauty to the scene. |
| | A medieval knight in shining armor, standing proudly in a lush, green field dotted with wildflowers, with a grand stone castle and rolling hills in the background. |
| | Several brightly colored rocks on a colorful beach, in the style of luminous spheres, emek golan, translucent color, 32k uhd, toyen, captivating |
| Figure D.17 | An East Asian girl with a simple wreath |
| | A Pacific Islander girl with a tropical flower crown, against a backdrop of a pristine beach at sunset, in a vibrant, colorful painting style. |
| Figure D.18 | Small cottage near the lake, snow, winter. |
| | Small cottage near the lake, summer. |
| Figure D.19 | A cinematic photo of cat [roubao] in space suit. |
| | A cinematic photo of cat [roubao] with flower. |
| | A cinematic photo of cat [roubao] in white chef suit. |
| | A cinematic photo of cat [roubao] in red outdoor jacket, Pixar anime style. |
| | A cinematic photo of cat [roubao] playing the piano, oil painitng style. |
| | A cinematic photo of cat [roubao] with black suit and sunglasses, on the beach. |

