# OpenReview forum: "UltraPixel: Advancing Ultra High-Resolution Image Synthesis to New Peaks"
_NeurIPS.cc/2024/Conference — NeurIPS 2024 poster_

### Official Review · Reviewer_UnLF · 2024-07-11

**Soundness:** 3
**Presentation:** 3
**Contribution:** 3
**Rating:** 8
**Confidence:** 4

**Summary:**

The paper presents UltraPixel, an innovative architecture for ultra-high-resolution image generation that tackles semantic planning, detail synthesis, and high resource demands. UltraPixel uses cascade diffusion models to generate images at multiple resolutions within a single model, efficiently guiding high-resolution generation with lower-resolution semantics. It features implicit neural representations for continuous upsampling and scale-aware normalization layers. Moreover, it requires less than a 3% increase for high-resolution outputs, boosting efficiency.

**Strengths:**

1、The paper demonstrates impressive results, with generated high-resolution images exhibiting remarkable detail. The proposed method outperforms existing approaches in terms of speed and flexibility, supporting arbitrary resolution image generation with a single model. This represents a significant advancement in the field.

2、The authors present a clear and well-motivated approach. They provide compelling evidence (Figures 2 and 6) to support their argument that the absence of low-resolution (LR) guidance can lead to suboptimal generation results.

**Weaknesses:**

1、 The manuscript's layout requires some refinement. For instance, Figure 4 extends beyond the page margins, and the text adjacent to Figure 9 appears overly condensed.

2、 Given that this is a text-to-image generation work, the paper would benefit from a more comprehensive set of visual results, including additional comparisons with state-of-the-art methods.

**Questions:**

1. The discussion in Section 4.3 regarding the timesteps of LR guidance extraction is intriguing. It would be valuable to see final generated images using different timesteps for guidance, rather than just attention visualizations.
2. The authors' use of LR guidance bears similarities to recent diffusion-based image super-resolution methods. A comparative discussion of these approaches could provide valuable context.
3. Given the method's design, it should theoretically support even higher resolutions (e.g., 5K, 6K). Have the authors explored this possibility?
4. The visual results demonstrating the limitations mentioned in the paper could be included in the supplementary materials to provide a more comprehensive understanding of the method's constraints.
5. Will the code and pre-trained models be made publicly available to facilitate reproducibility and further research in this area？

**Limitations:**

See Weaknesses.

---

> ### Author Rebuttal · Authors · 2024-08-07
>
> `Q1`: **The manuscript's layout requires some refinement.**
>
> Thank you for your constructive suggestion. We have carefully revised the format accordingly.
>
> `Q2`: **Paper would benefit from a more comprehensive set of visual results, including additional comparisons with state-of-the-art methods.**
>
> Thank you for the valuable advice. In the rebuttal PDF, we provide visual comparisons with diffusion-based SR methods, including SUPIR and StableSR in **Figure 1**, and leading T2I commercial products, DALLE 3 and MidJourney V6 in **Figure 4**. Our results are visually pleasing and richer in detail. We have included more comparisons in the revised supplementary materials.
>
> `Q3`: **Final generated images using different timesteps for guidance**
>
> Thank you for the constructive advice. We include visual results of the ablation study on LR guidance timesteps in **Figure 2** of the rebuttal PDF. The figure illustrates the LR generation process and HR generation processes with different guidances, considering three cases: **(1) ‘t=t^H’** where the timestep of LR guidance is synchronized with the timestep of HR generation, **(2) ‘t=0.5T’** means LR features at the middle timestep, and **(3) ‘t=0.05T’** near the end. The HR generation process establishes the overall semantic plan in the early stage. However, synchronized or middle timestep guidance is too noisy to provide sufficient direction at this stage. In contrast, the final step of LR guidance delivers clear semantic information. With semantics-rich guidance, the HR image generation can produce coherent structures and fine-grained details. We have included this visualization in the revised manuscript.
>
> `Q4`: **Comparative discussion of diffusion-based image SR methods**
>
> Thank you for the valuable suggestion. As a T2I task, our objective is to generate a high-quality, high-resolution image aligned with the text input, rather than merely enlarging the synthesized low-resolution (LR) image. Unlike SR methods that typically use LR images as input, we adopt multi-level intermediate model features as LR guidance for high-resolution generation. This approach allows for refinement when the LR guidance is unreliable.
>
> We compare our method with state-of-the-art diffusion-based SR methods, namely **SUPIR** and **StableSR**. The visual results in **Figure 1** of the rebuttal PDF demonstrate that our method produces more reasonable structures and richer details. Notably, our method excels in refining images that contain artifacts in the LR version, enhancing visual quality, particularly in facial and hand details, as shown in the second example. Additionally, processing a 4096x4096 image takes **12 minutes** for SUPIR and **11 minutes** for StableSR on an A100 GPU, while our model requires only approximately **78 seconds**.
>
> We also evaluate the FID, patch-based FID, Inception, Inception-score, PickScore, clip-score at resolution of 4096x4096 , as shown below, where our method achieves superior results. These findings collectively underscore the effectiveness of our approach. We have included these comparisons in the revised manuscript.
>
>   method   | PickScore (vs StableSR) $\uparrow$ |PickScore (vs SUPIR)$\uparrow$| FID$\downarrow$| FID_p$\downarrow$| IS$\uparrow$| IS_p$\uparrow$| CLIP$\uparrow$| Latency(sec.)$\downarrow$|
> |---|---|---|---|---|---|---|---|---|
> | StableSR           | 31.3%    | - | 65.27| 48.18|27.55|9.25|32.49|728
> | SUPIR            | -    | 34.7%   |64.13|46.98|26.16|9.83|31.28|682
> | Ours | **68.7%**  | **65.3%**|**63.80**|**44.32**|**27.65**|**10.23**|**33.10**|**78**
>
> `Q5`: **Support even high resolution image generation**
>
> Yes, we have further trained a model that supports up to 6K generation. Some visual results at resolutions of **5760x3840** and **3072x6144** are provided in **Figure 8** of the rebuttal PDF. We have included more visual results in the revised manuscript and will make the model publicly available.
>
> `Q6`: **The visual results demonstrating the limitations mentioned in the paper could be included in the supplementary materials to provide a more comprehensive understanding of the method's constraints**
>
> Thank you for the constructive advice. We found that our model may occasionally fail in generating accurate structures in people's hands. Introducing more human hand data should help address this issue. We will include a more detailed analysis of both the strengths and weaknesses of our model in the revised paper.
>
> `Q7`: **Will the code and pre-trained models be made publicly available to facilitate reproducibility and further research in this area**
>
> Yes, all models and code will be released to promote the development of the community.

---

> > ### Comment · Reviewer_UnLF · 2024-08-09
> >
> > The author's rebuttal addresses my concerns. The explanation using a figure is clear. After reading the rebuttal and other reviewers' comments, I think this work is worth being accepted for subsequent researchers to follow up on their studies.

---

### Official Review · Reviewer_z6xV · 2024-07-12

**Soundness:** 3
**Presentation:** 3
**Contribution:** 3
**Rating:** 6
**Confidence:** 4

**Summary:**

This paper introduces UltraPixel, a method for generating high-quality ultra-high-resolution images. It utilizes the semantics-rich representations of lower-resolution images in a later denoising stage to guide the overall generation of highly detailed high-resolution images. The method incorporates implicit neural representations for continuous up-sampling and scale-aware normalization layers that are adaptable to various resolutions. The experimental results show that it has excellent ability in generating high-resolution images of different sizes.

**Strengths:**

1. The introduction of implicit neural representations for continuous up-sampling and scale-aware normalization layers adaptable to various resolutions is a creative solution that addresses a challenge in the scalability of image generation models.

2. The methodology is well-articulated, with a clear explanation of how the model manages to generate high-quality images while maintaining computational efficiency.

3. The ablation experiments are thoroughly conducted, systematically revealing the contribution of each component to the overall performance.

4. The paper proposes an innovative method for generating high-quality, ultra-high-resolution images efficiently, tackling a major challenge in image synthesis.

**Weaknesses:**

1. The explanation of the implicit neural representation (INR) requires further clarity regarding its ability to enable continuous upscaling. Moreover, an in-depth analysis and dedicated ablation study of the Scale-Aware Normalization (SAN) feature would provide insights into its role in resolution adaptability.

2. To underscore the advantages of the proposed framework, the experiments should be expanded to include comparative analyses with Latent Diffusion Model (LDM)-based and pixel-based image synthesis methods, showcasing the superior performance of the framework in high-resolution image generation tasks.

**Questions:**

1. Why is the perturbed version $𝑧_{1}$ preferred over the $𝑧_{0}$ from the low-resolution synthesis for guidance purposes?

2. Regarding Figure 4, could you clarify why additional learnable tokens are integrated with the guidance tokens for the self-attention mechanism, instead of solely relying on the guidance tokens? What unique function do these learnable tokens serve?

3. Can you outline the computational steps involved in the implicit neural representation? Is there a need for manually specifying positions?

4. What justifies the forms of Equations (3) and (4), which amalgamate terms with distinct physical interpretations? Is there an underlying principle that supports their direct summation, as it seems to go against intuitive reasoning?

5. In the context of line 255, the use of 𝑡=0.5 and 𝑡=0.05 is ambiguous. Are these intended to denote specific sampling stages within the low-resolution synthesis—fixed and terminal steps, respectively? Consequently, is 𝑡=1 encompassed within the scenario where 𝑡=0.5?

**Limitations:**

1. The paper may not sufficiently address how well the model generalizes to datasets beyond the training distribution. It is crucial to understand if the model's performance degrades with different or less common image content.

2. There is a need for more rigorous testing of the model's robustness to various corruptions and perturbations that could be encountered in real-world applications.

---

> ### Author Rebuttal · Authors · 2024-08-07
>
> `Q1` **Clarification on INR and analysis of SAN feature**
>
> Compared to discrete grid pixels, INR represents data as a neural function, mapping continuous coordinates to signals. Its representation capacity depends not on grid resolution but on the neural network's ability to capture underlying data structures, reducing redundancy and providing a compact yet powerful consistent representation. INR has proven useful in various 3D/2D tasks, such as NeRF[1] and LIIF[2]. To illustrate the effectiveness of INR , we provide visual examples in Fig 6 of the rebuttal PDF. With INR guidance, our model consistently generates high-quality images across different resolutions. In contrast, using simple bilinear upsampling followed by several conv. fails to provide clear guidance for higher resolution images like 4K, resulting in noisy artifacts. The quantitative comparisons of ‘BI+Conv’ and ‘INR’ in Table 3 of the main paper also support this observation.
>
> Regarding SAN, we compute feature statistics for varying resolutions, with and without SAN, as shown in Fig. 5 of the rebuttal PDF. With SAN, different resolutions have similar feature distributions, while without SAN, mean values vary significantly. This shows SAN enables stable handling of different resolutions, resulting in better visuals.
>
> [1] Nerf: Representing scenes as neural radiance fields for view synthesis. ECCV 2020
>
> [2] Learning continuous image representation with local implicit image function. CVPR 2021
>
> `Q2` **Compare with LDM-based and pixel-based methods**
>
> Our method is fundamentally a LDM model, performing the generative diffusion process in a latent space. The methods we compared, including Demofusion, ScaleCrafter, FouriScale, and Pixart-Sigma, are all LDM adapted with various techniques for HR image generation. To our best knowledge, no open-source pixel-based synthesis method can directly generate HR images due to the extreme memory and computational demands of doing so in pixel space.
>
> To demonstrate our method's superiority, we compare it with leading T2I products, DALLE-3 and MJ V6. Quantitative evaluation shows our method is favored by PickScore in 70% of cases against DALLE-3. Visual comparisons in Fig 4 of the rebuttal PDF show our images are of comparable quality to both products.
>
> `Q3` **Why z1 over z0**
>
> We do not use z1 or z0 latents. Instead, we utilize intermediate multi-level model features by forwarding z1 to the base model.
>
> `Q4` **Why additional learnable tokens**
>
> Unlike LR guidance tokens with a strong local bias, the learnable tokens can globally query and aggregate compact useful information from guidance tokens, enhancing the model’s comprehension  capabilities. Additionally, learnable tokens have proven effective in adapting to various vision tasks, as evidenced by  works [3-5]. While the LR guidance tokens primarily encode local patterns, the learnable tokens acquire auxiliary information from large datasets, improving the adaptability and performance of models.
>
> [3] Vision transformers need registers. ICLR 2024
>
> [4] End-to-end object detection with transformers. ECCV 2020
>
> [5] Generalized decoding for pixel, image, and language. CVPR 2023
>
> `Q5` **Steps involved in the INR**
>
> After obtaining the LR feature, INR is involved in all HR generation sampling steps to forward the LR guidance to the HR branch.
>
> `Q6` **What justify eq 3 4**
>
> Eq (3) (4) inject time and image scale information into the model using AdaIN, which predicts scale and shift from the input information to modulate model features. This technique is widely employed and proven effective in injecting style or time information into generative models [6-8].
>
> [6] High-resolution image synthesis with latent diffusion models. CVPR 2022
>
> [7] Sdxl: Improving latent diffusion models for high-resolution image synthesis. ICLR 2024
>
> [8] A style-based generator architecture for generative adversarial networks. CVPR 2019
>
> `Q7` **Clarification of t**
>
> We have revised the notation to t=0.5T and t=0.05T. For example, when the number of sampling steps is 20 (T=20), t=0.5T refers to using the LR features at the 10th step to guide all HR sampling steps. As shown in Table 4 of the main paper and Fig 3 of the rebuttal PDF, t=0.05T is preferred, as the LR features at this time point provide clear semantic guidance for HR generation. We do not consider t=T, as the LR features at the first step of the LR sampling stage are too noisy to provide useful information.
>
> `Q8` **Generalization concern**
>
> As mentioned in lines 177-178, we build our model upon the well-trained 1024^2 StableCascade, which is trained on huge datasets and generalizes well across various scenarios. As described in lines 10-13 and 170-172, we train an additional 3% of the parameters specifically for high-resolution image generation, while keeping the other parameters frozen. This method preserves the model's generative power. As shown in Fig 1 of the main paper, and Fig 1 (“A photo of an astronaut riding a horse in the forest. There is a river in front of them with water lilies.”) and 2 (“Dogs sitting around a poker table”) in the rebuttal PDF, our method can generate less common content of high quality. Fig 8 in the rebuttal PDF also shows that our method can generate pleasing images of various styles (photo-realistic or oil-painting) and content (real or imaginary).
>
> `Q9` **Model robustness**
>
>  Our model performs well in situations involving corruptions and perturbations. Fig 7 of the rebuttal PDF shows that even when the prompts are miswritten as "2014 brabus b63s 700 6x6 mercedes benz g class hd pictures" and " Ext for in ldg and sc gatlinburg cabin wahoo sale cabins rentals of american homes tn log city" our method accurately generates high-quality 4K images. Even if the prompt is incomplete, jumbled, or contains spelling errors, our method still can generate high-quality images. This demonstrates the robustness of our method in understanding and interpreting diverse and imperfect input.

---

> ### Author Response · Authors · 2024-08-13
> **Response to z6xV by Authors**
>
> We would like to thank you again for the valuable time you devoted to reviewing our paper. Since the end of discussion period is getting close and we **have not heard back from you yet**, we would appreciate it if you kindly let us know of any other concerns you may have, and if we can be of any further assistance in clarifying them.
> Thank you once again for your contribution to our paper's refinement.

---

### Official Review · Reviewer_rwnV · 2024-07-13

**Soundness:** 2
**Presentation:** 3
**Contribution:** 3
**Rating:** 4
**Confidence:** 5

**Summary:**

This paper presents a method for Ultra-High-Resolution image generation from text prompts. The method is based on StableCascade. The original StableCascade can generate 1024x1024 images. This paper proposes another HR latent diffusion model that can utilize the guidance from 1024 x 1024 images and generate 4096 x 4096 images. Unlike previous methods that directly use the low-resolution output, the method chooses to use the features of the base model as guidance and proposes an implicit-based method to upsample the low-res guidance features.

**Strengths:**

- The idea of guidance feature and implicit-based upsampling is simple but effective.
- The paper reads well, and the presentation is clear.
- The results are very impressive.
- The proposed method only needs light-weight finetuning from StableCascade.

**Weaknesses:**

- More validation and analysis are needed. In the comparison, a traditional image upsampler is used, but the traditional image upsampler is often smaller and also trained on much smaller datasets. For a fair comparison, it will be good to compare with the state-of-the-art generative image upsampler such as StableSR and Stable Diffusion Upscaler.
- A comparison of this baseline is missing: instead of using guidance features, the HR latent model can directly use the LR images / latents from the base model.
- It would be good to have visual results of the ablation on LR guidance timesteps.
- Ablation on scale-aware normalization is missing.

**Questions:**

- Is the base model frozen from StableCascade?
- Is the implicit model jointly trained with the HR latent model?

**Limitations:**

It will be good to show some visual failure cases.

---

> ### Author Rebuttal · Authors · 2024-08-07
>
> `Q1` **Comparison with the SOTA generative upsampler**
>
> Thank you for the valuable suggestion. We compare our method with SOTA diffusion-based SR methods, namely **SUPIR** and **StableSR**. The visual results in **Figure 1** of the rebuttal PDF demonstrate that our method produces more reasonable structures and richer details. As a T2I task, our objective is to generate a high-quality, high-resolution image aligned with the text input, rather than merely enlarging the synthesized low-resolution (LR) image. Notably, our method excels in refining images that contain artifacts in the LR version, enhancing visual quality, e.g., facial and hand details in 2nd example. Besides, processing a 4096x4096 image takes **12 minutes** for SUPIR and **11 minutes** for StableSR on an A100 GPU, while our model requires only approximately **78 seconds**. We also evaluate the PickScore, FID, patch-based FID,Inception-score (IS), patch-based IS,  CLIP-score on resolution of 4096x4096, as shown below, where our method achieves superior results. These findings collectively underscore the effectiveness of our approach. We have added these comparisons in the revised manuscript.
> |       method   | PickScore (vs StableSR) $\uparrow$ |PickScore (vs SUPIR)$\uparrow$| FID$\downarrow$| FID_p$\downarrow$| IS$\uparrow$| IS_p$\uparrow$| CLIP$\uparrow$| Latency(sec.)$\downarrow$|
> |---|---|---|---|---|---|---|---|---|
> | StableSR           | 31.3%    | - | 65.27| 48.18|27.55|9.25|32.49|728
> | SUPIR            | -    | 34.7%   |64.13|46.98|26.16|9.83|31.28|682
> | Ours | **68.7%**  | **65.3%**|**63.80**|**44.32**|**27.65**|**10.23**|**33.10**|**78**
>
> `Q2` **Comparison with the baseline using latent of base model**
>
> As suggested, we quantitatively and qualitatively compare our UltraPixel method with the baseline that **directly uses the low-resolution (LR) latent generated by the base model (an SR pipeline)**. We evaluate the PickScore, FID, patch-based FID,Inception-score (IS), patch-based IS,  CLIP-score on resolution of 2048x2048 in the table below. As shown in the table below, our method consistently outperforms the baseline across all metrics, achieving a **71.1%** win-rate on PickScore . **Figure 3** in the rebuttal PDF illustrates that using LR latents causes the model to overly rely on the LR input, often failing to refine details further. In contrast, our approach, which utilizes multi-level semantics-rich features, provides  semantic-rich guidance and allows for additional refinement. Consequently, our method produces higher-quality images with more reasonable structures and richer details. We have included these details in the revised manuscript.
> |       method   | PickScore $\uparrow$| FID$\downarrow$| FID_p$\downarrow$| IS$\uparrow$| IS_p$\uparrow$| CLIP$\uparrow$
> |---|---|---|---|---|---|---
> | Baseline | 28.9%   |63.32|48.19|26.73|11.88|30.38
> | Ours | **71.1%**  |**62.82**|**43.97**|**29.67**|**14.15**|**33.18**
>
> `Q3` **Visual results of the ablation on LR guidance timesteps**
>
>  Thank you for the constructive advice. We include visual results of the ablation study on LR guidance timesteps in **Figure 2** of the rebuttal PDF. The figure illustrates the LR generation process and HR generation processes with different guidances, considering three cases: **(1) ‘t=t^H’** where the timestep of LR guidance is synchronized with the timestep of HR generation, **(2) ‘t=0.5T’** means LR features at the middle timestep, and **(3) ‘t=0.05T’** near the end. The HR generation process establishes the overall semantic plan in the early stage of the sampling process. However, synchronized or middle timestep guidance is too noisy to provide sufficient direction at this stage. In contrast, the final step of LR guidance delivers clear semantic information. With semantics-rich guidance, the HR image generation can produce coherent structures and fine-grained details. We have included this visualization in the revised manuscript.
>
> `Q4` **Ablation on scale-aware normalization**
>
> We have quantitatively evaluated the performance of our model with and without scale-aware normalization (SAN) in **Table 3 of the main paper**. The method labeled ‘INR’ refers to the model without SAN, while ‘INR+SAN’ incorporates SAN. The results indicate that SAN consistently enhances performance across different resolutions. Additionally, we compute feature statistics (mean and variance) for varying resolutions, with and without SAN, as shown in **Figure 5** of the rebuttal PDF. With SAN, the features of different resolutions exhibit similar distributions, while the mean values vary significantly across resolutions without SAN. This demonstrates that SAN enables our model to handle different resolutions stably, resulting in better visual results.
>
> `Q5` **Is the base model frozen from StableCascade**
>
> Yes. As described in lines **170-172**, the base model is frozen
>
> `Q6` **Is the base model frozen from StableCascade**
>
> Yes. As described in lines **170-172**, the implicit model is jointly trained with the HR branch.

---

> ### Author Response · Authors · 2024-08-13
> **Response to rwnV by Authors**
>
> We would like to thank you again for the valuable time you devoted to reviewing our paper. Since the end of discussion period is getting close and we **have not heard back from you yet**, we would appreciate it if you kindly let us know of any other concerns you may have, and if we can be of any further assistance in clarifying them.
> Thank you once again for your contribution to our paper's refinement.

---

### Author Rebuttal · Authors · 2024-08-07

**Response to AC and reviewers (with PDF)**


We sincerely appreciate your time and efforts in reviewing our paper. We are glad to find that reviewers recognized the following merits of our work:
- **Innovative and effective solution [rwnV, z6xV, UnLF]**: The proposed UltraPixel introduces a Low-Resolution (LR) guidance feature to reduce the complexity of High-Resolution (HR) image generation. Additionally, it employs an implicit function and scale-aware normalization to assist the network in generating images of varying resolutions. This approach is both innovative and novel.
- **Impressive results [rwnV, z6xV, UnLF]**: UltraPixel generates ultra-high-resolution images with impressive quality and rich details, effectively addressing a major challenge in image synthesis.
- **Clarity and Readability [rwnV, z6xV, UnLF]**: Our paper is well-motivated, clearly articulated, and easy to read.

We also thank all reviewers for their insightful and constructive suggestions, which help further improve our paper. In addition to the pointwise responses below, we summarize the major revision in the rebuttal according to the reviewers’ suggestions

- **Comparative study on advanced T2I and diffusion-based super-resolution methods**: We have incorporated extensive quantitative and qualitative comparisons with leading commercial T2I products, DALLE 3 and MidJourney V6, as well as state-of-the-art SR methods, SUPIR and StableSR. These results further demonstrate UltraPixel's impressive ability to generate ultra-high-resolution images and its superior efficiency.
- **Enhanced visual analysis and method clarification**: We have added visual comparisons of different timesteps for extracting LR guidance, feature distribution across different resolutions, and an in-depth analysis of the proposed method.

Best,

Authors

---

### Decision · Program_Chairs · 2024-09-25

**Decision:**

Accept (poster)

**Comment:**

The paper received mixed ratings of Strong accept, Weak accept, and a Weak reject. The major strength of the paper lies in its effective approach to high-resolution photorealistic image synthesis, impressive results as shown extensively in the paper, and a well-written paper.

The negative reviewer has several concerns regarding the experimental validation. More specifically, missing comparisons with the state-of-the-art upsampler, baseline architecture (e.g., directly using LR image/latent), and ablation on LR guidance timesteps.

The authors provide a rebuttal, including additional experimental results that address these concerns. However, the reviewer did not engage with the rebuttal. Two other reviewers are satisfied with the rebuttal and agree that the paper has sufficient merits.

There is also an ethic review raising issues with self-collected image datasets. The authors' rebuttal clarified the source of the dataset (Unsplash Full Dataset), potential fairness issues and mitigation strategies, and potential negative societal impacts.

Based on the reviews and the author's rebuttal, the AC believes that the concerns have been clarified and recommends to accept.